# Impact of pulsed-wave-Doppler velocity-envelope tracing techniques on classification of complete fetal cardiac cycles

Eleonora Sulas[1]*, Emanuele Ortu[1], Monica Urru[2], Roberto Tumbarello[2], Luigi Raffo[1], Giuliana Solinas[3], Danilo Pani[1]

1 Department of Electrical and Electronic Engineering, University of Cagliari, Cagliari, Italy, 2 Division of Pediatric Cardiology, San Michele Hospital, Cagliari, Italy, 3 Department of Biomedical Science, University of Sassari, Sassari, Italy

* sulaseleonora1992@gmail.com

**Data Availability Statement:** The data is available through the Mendeley data at DOI: 10.17632/pg2djw3sjc.1.

## Abstract

Fetal echocardiography is an operator-dependent examination technique requiring a high level of expertise. Pulsed-wave Doppler (PWD) is often used as a reference for the mechanical activity of the heart, from which several quantitative parameters can be extracted. These aspects suggest the development of software tools that can reliably identify complete and clinically meaningful fetal cardiac cycles that can enable their automatic measurement. Several scientific works have addressed the tracing of the PWD velocity envelope. In this work, we assess the different steps involved in the signal processing chains that enable PWD envelope tracing. We apply a supervised classifier trained on envelopes traced by different signal processing chains for distinguishing complete and measurable PWD heartbeats from incomplete or malformed ones, which makes it possible to determine the impact of each of the different processing steps on the detection accuracy. In this study, we collected 43 images and labeled 174,319 PWD segments from 25 pregnant women volunteers. By considering seven envelope tracing techniques and the 23 different processing steps involved in their implementation, the results of our study reveal that, compared to the steps investigated in most other works, those that achieve binarisation and envelope extraction are significantly more important ($p < 0.05$). The best approaches among those studied enabled greater than 98% accuracy on our large manually annotated dataset.

## Introduction

Ultrasonography is the leading technology for the diagnosis and monitoring of fetal heart pathologies [1]. Among the available modalities, fetal echocardiography typically takes advantage of pulsed-wave Doppler (PWD) for the assessment of the cardiac rhythm [2, 3], which enables objective analysis of the blood flow through the heart. In particular, in routine Doppler examinations, the apical view is considered to be the best for fetal heart inspection in cases of suspected valvular heart disease [4]. Using the apical five-chamber view, the four cardiac chambers and the first part of the aorta (assumed to be a fifth chamber) can be investigated.

**Funding:** Eleonora Sulas is grateful to Sardinia Regional Government for supporting her PhD scholarship (P.O.R. F.S.E., European Social Fund 2014-2020). Part of this research was supported by the Italian Government–Progetti di Interesse Nazionale (PRIN) under the grant agreement 2017RR5EW3 - ICT4MOMs project.

**Competing interests:** The authors have declared that no competing interests exist.

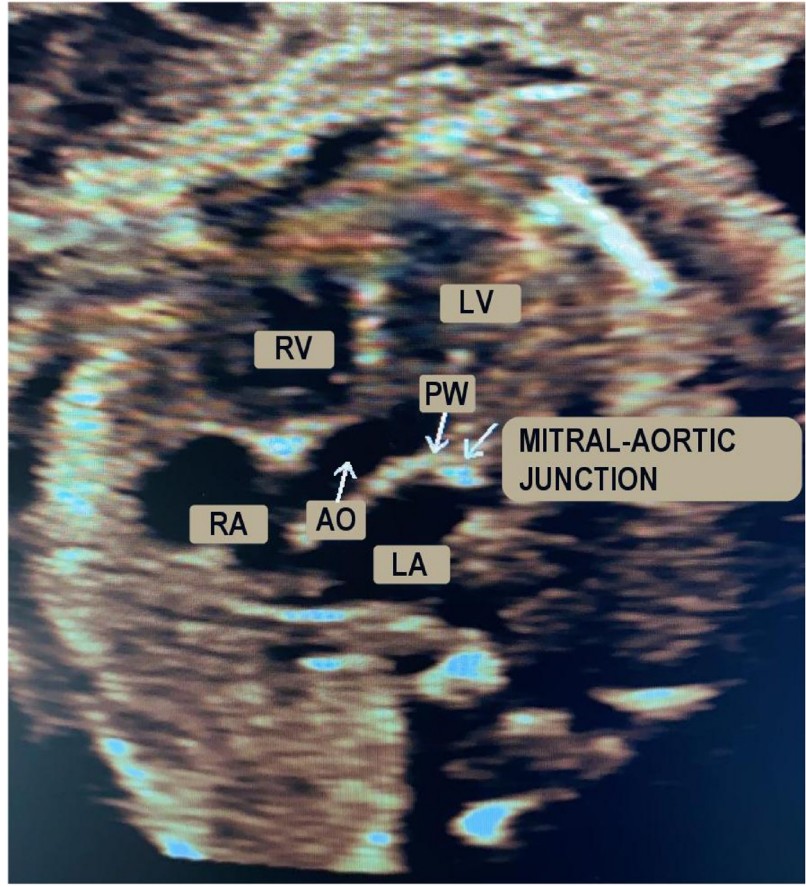

**Fig 1. Apical five-chamber B-mode view of the fetal heart.** In detail: right ventricle (RV), right atrium (RA), left ventricle (LV), left atrium (LA), aortic region (AO) and sample volume positioning (PW).

Fig 1 shows an apical five-chamber view in which the two atria, two ventricles and the aortic region are visible and labeled. By positioning the sample volume on the mitral-aortic junction (PW in Fig 1), the morphologies of the diastolic and systolic functions are clear and recognizable in the PWD spectrum [5], as illustrated in Fig 2. Overall, there are two main waves: the so-called E/A wave, where E is the first peak and A is the second, and the V wave. The E/A wave refers to the mitral inflow and the V wave represents the aortic outflow. These two waves have opposite polarity. Specifically, the E peak (E for "early") refers to the early, passive diastolic filling of the ventricle, which depends on the ventricular wall relaxation, and the A peak (A for "atrial") is the wave associated with the active filling of the ventricle [6]. E/A wave abnormality is an indication of systolic dysfunction. The V peak (V for "ventricular") is the wave associated with the ventricular systole.

By representing the blood flow when the sample volume is positioned on the mitral-aortic junction, the PWD provides a good reference signal for the mechanical heart activity during systole and diastole. Several clinical indexes can be extracted from this signal, the fetal heart rate being the simplest. Different guidelines suggest baseline values ranging between 110 and 150-160 bpm before and during labor [7]. The E/A ratio is an index for the evaluation of the left ventricle of the heart, where E is smaller than A, and the E/A ratio increases during pregnancy towards a value of one. After birth, the E/A ratio becomes greater than one [8]. During early pregnancy (8-20 weeks), anemic fetuses present a dominant E peak, which suggests that

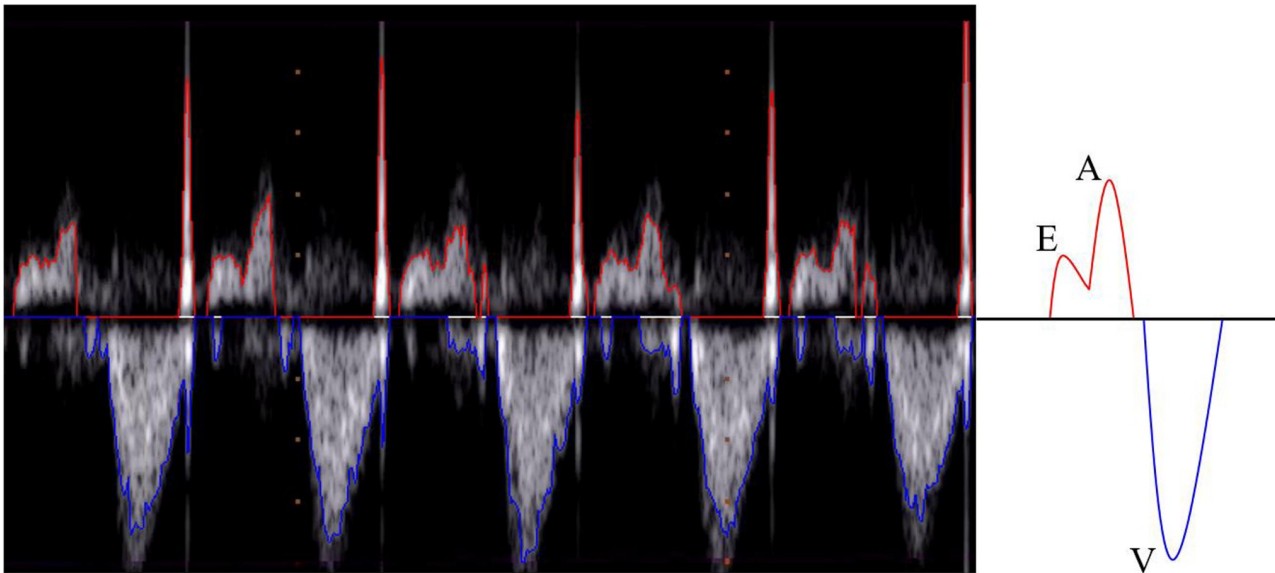

**Fig 2. Pulsed-wave Doppler image.** (Left) The initial segment of the image shows the first five beats from the first PWD image in our shared dataset. The image includes the upper (red line) and lower (blue line) envelopes. (Right) A graphical representation of the ideal waveform morphology of the envelope associated with a fetal beat, which was acquired using PWD with the apical five-chamber window.

in fetal anemia, there is an increased preload in the right atrium [9]. The atrio-ventricular interval (AV) is an important index of conduction disorders that can be derived from the PWD. In fact, a normal heart rate with a prolonged AV conduction time interval may progress to a complete AV block. Measurement of the mechanical AV conduction time interval could aid in the identification of fetuses with first-degree AV block. These fetuses might benefit from transplacental steroid administration to prevent progression toward complete and irreversible AV block [10]. The myocardial performance index was reported by Tei [11] to be a useful predictor of systolic and diastolic cardiac dysfunction. This index is defined as the sum of the isovolumetric contraction time and isovolumetric relaxation time, divided by the ejection time from each ventricle. These time indexes are also very useful indicators of fetal wellbeing [12]. The mean velocity is also used for the assessment of Doppler images [13].

Despite the possibility of obtaining such a large amount of information from this signal, the fetal PWD procedure is challenging for cardiologists to perform and, like any other antenatal ultrasound examination, it is strongly operator-dependent and requires a high level of expertise. To support ultrasound operators with limited skills to perform fetal echocardiography, beyond tele-mentoring sessions [14], or for the automatic analysis of long traces, it is important to develop automatic tools for the detection of complete and clinically meaningful fetal heartbeats in the PWD trace [15]. Once identified, these heartbeats can be analyzed for extraction of the aforementioned clinical parameters and quantification of the cardiac function.

Recent works have addressed this important issue using ad-hoc algorithms and machine learning [15–17]. However, to the best of our knowledge, there has been no systematic analysis of the steps comprising the signal processing chain for tracing the PWD velocity envelope. Such analysis is necessary to support the development of effective methods that enable the improvement of steps that are critical for the effective detection and processing of clinically valuable fetal heartbeats in the PWD image. Compared to the use of synthetic indexes such as the mean velocity or other parameters that can be extracted from the Doppler image, by

focusing only on the PWD envelope it is possible to reduce the occurrence of computational errors. Moreover, this focus is closely aligned with the usual clinical experience in the identification of well-formed PWD traces that have clinical value for subsequent analyses.

On this basis, and after an in-depth analysis of the scientific literature regarding methods for the automatic tracing of the velocity envelope from Doppler images, in this work, we identified the main processing steps adopted in Doppler envelope tracing. These steps were then combined to evaluate their relative importance in defining a near-optimal processing chain for the extraction of features to feed a supervised classifier trained to distinguish fetal heartbeats that are complete and measurable from those that are incomplete or malformed. As such, as the objective is not to accurately trace PWD envelopes, but rather to identify clinically meaningful fetal heartbeats in the PWD signal, the performance evaluation was conducted using traditional machine learning indexes rather than comparing manual tracings. From this perspective, the optimization of the classifier is of minor interest, considering its performance as a bias for the assessment of all the PWD tracing methods. Validation of the proposed approach was performed on a large custom dataset of fetal PWD recordings, which is publicly accessible with this article.

## Related works

The automatic and unsupervised detection of meaningful fetal cardiac cycles in Doppler ultrasound videos was originally presented in [17], and subsequent development of the original technique is presented in [15, 16]. To the best of our knowledge, no other works have focused on the automatic detection of complete and clinically measurable fetal cardiac cycles in a Doppler ultrasound trace. In the cited works, detection was performed on PWD signals acquired using an apical five-chamber window, which were characterized by two envelopes: one for the diastolic phase and the other for the systolic phase. The idea behind these methods was to reduce the complexity of the classification by using the envelopes rather than the PWD image.

This approach is dependent on the adopted envelope tracing method and very different methods have been developed to achieve this aim. In this study, we identified these methods and systematically assessed the impact of their individual processing steps on the final classification outcome. This section briefly introduces the techniques used and developed for envelope tracing in the contexts envisioned by the authors. In the Materials and methods section, we provide a detailed technical description of the processing steps studied in this work. Table 1 lists the main processing steps adopted in studies focused on the automatic tracing of Doppler envelopes [16, 18–23].

Ref. [18] introduced the automated analysis of a general Doppler ultrasound velocity flow and presented a method comprising the following processing steps: (i) image filtering, using a Gaussian-shaped low-pass filter with a $\sigma = 1.5$ to remove high-frequency noise, (ii) edge detection by a non-linear Laplace edge detector (NLLAP), (iii) suppression of spurious edges, and (iv) extraction of the overall envelope function using a custom algorithm known as the *biggest-gap* algorithm, which is with the other processing steps in the next section. The authors compared the envelope extraction results with manual tracings and determined their concordance using Bland-Altman analysis.

In [19], the authors studied patients with atrial fibrillation. Using the same process described in [18], they focused more on contrast enhancement, whereby just after the application of the Gaussian kernel to remove background noise and the stretch of the contrast, they used a k-means algorithm to organize the pixel intensities of the image into three main clusters, representing weak-signal, strong-signal, and background pixels. The centers of these

**Table 1. Main publications on the automated tracing of the Doppler velocity envelope.**

| | Image preprocessing 1 | Image preprocessing 2 | Image binarisation | Image postprocessing | Envelope extraction | Envelope post processing |
|---|---|---|---|---|---|---|
| [18] | Gaussian-shaped low-pass filter with $\sigma = 1.5$ | | Modified non-linear Laplace edge detector | Region-growing algorithm | Biggest-gap algorithm | five-point averaging filter |
| [19] | Gaussian-shaped $5 \times 5$ low-pass filter with a $\sigma = 1.5$ | k-means algorithm to enhance the contrast | Combination of modified non-linear Laplace and Sobel edge detectors | Removal of spurious areas (unspecified number of connected pixels) | Biggest-gap algorithm | |
| [20] | | | Otsu Threshold | Removal of spurious areas with $<$50 connected pixels + Dilation step | Trace the boundary pixels of all white regions to get contours | Median filter with 3 taps, cut-off frequency $\approx 70$ Hz |
| [21] | | | Local adaptive threshold | | Envelope extraction [†] | Median filter with 15 taps |
| [22] | Gaussian kernel of $5 \times 5$ with a $\sigma = 1.5$ | Intensity adjustment by images subtraction | Canny algorithm ($\sigma = 0.5$) | | Peak velocity extraction [†] | |
| [23] | | | Steepest gradient histogram threshold | Removal of small noisy clusters with $<$ 500 connected pixels | Biggest-gap algorithm | low-pass first-order Butterworth. |
| [16] | | | Otsu 2D method | Removal of small noisy clusters with $<$70 4-connected pixels | Trace the boundary pixels of all white regions to obtain contours | |

[†]: This method was not specified in the original manuscript, so that adopted in [16] was used.

three clusters were applied to find two optimal thresholds to enhance important signal information in the image. To facilitate annotation of the heartbeats in the Doppler traces, the recordings included the ECG signal. After tracing the envelope, the authors fitted a parametric model to each cardiac cycle, based on the sum of cosines. They were able to apply these models to the cardiac cycles because of the presence of the fetal heartbeat annotations. The results were shown in terms of the errors between the clinical parameters computed on the automatically traced envelope and those manually traced. Beat-by-beat and average beat comparisons of the two were performed, with the relative errors determined by Bland-Altman analysis.

Later, Ref. [20] focused on retrieval of the shape-based similarities of Doppler images to support clinical decision-making. The authors binarized the images based on an Otsu threshold, followed by dilation and then extraction of the envelope, which was traced by considering the boundary pixels of the white regions. A three-tap temporal median filter was then applied to smooth the envelope.

The authors in [21] adopted a two-dimensional (2D) Doppler echocardiography for the non-invasive automated assessment of the severity of aortic regurgitation. They used a local adaptive threshold to binarize images, filtered the envelope using a median filter with 15 taps, and then interpolated the envelope using a cubic spline to achieve a sampling frequency of 1 kHz. This method also relied on ECG gating to identify the required fiducial points. The results were then compared with manually-traced envelopes by Bland-Altman analysis.

The main aim of the study reported in [22] was again the automated assessment of aortic regurgitation using 2D Doppler echocardiography. The applied algorithm in that study was decomposed into the following steps: (i) application of a $5 \times 5$ Gaussian kernel with $\sigma = 1.5$, (ii) conversion of the filtered image into a grayscale image, (iii) morphological operation on the image by disk approximation, (iv) subtraction of the images obtained in the first and second steps to adjust the intensity, and (v) Canny edge detection ($\sigma = 0.5$). These results were also compared with a manually-traced envelope.

The study in [23] analysed 30-second adult Doppler traces acquired using an apical five-chamber window positioned in the left ventricular outflow tract. First, the authors identified the Doppler region of interest and converted it from RGB to grayscale format. To filter out small noisy clusters, they removed connected areas with fewer than a predefined number of pixels (empirically chosen as 500 pixels). The maximum velocity profile was then extracted from the resulting filtered image using the biggest-gap method. To filter out high-frequency noise, they applied a low-pass first-order Butterworth digital filter to the initial velocity profile. ECG data were collected simultaneously from a subgroup of patients with heart rates of 60-96 bpm. In the analysis of the Doppler envelopes in the left ventricular outflow tract, based on the typical monophasic cardiac cycle pattern, each cardiac cycle was considered to have two base points, one at the onset and the other at the end of the curve, as well as a peak. Peak points on the smoothed velocity profile were identified by imposing a constraint whereby the distance between two consecutive peaks should be not smaller than 80% of the duration of the cardiac cycle. To determine the location of the pattern base points, the authors computed the first derivative of the velocity curve. Lastly, to obtain the final automated traces, they fitted a third-order Gaussian model to each cardiac cycle of the velocity profile.

## Materials and methods

In this section, we provide a technical description of the processing steps of the algorithms identified in the previous section, which were used in our systematic analysis presented here, even when they are outside the context of the original algorithms. We then describe the assessment process and dataset adopted in this investigation.

By analyzing the most important aspects related to envelope tracing, with guidance from the studies briefly reviewed in the section above, we were able to recognize five main phases: (i) image preprocessing, (ii) image binarization, (iii) image post-processing, (iv) envelope extraction, and (v) envelope post-processing. Considering the processing steps used in previous works (Table 1), we found that the first phase can be also composed of two separate consecutive processes. For this reason, in the following, six steps are considered and analyzed. Fig 3 shows the complete workflow, including the different options for the various processing steps used in the literature for envelope tracing, which are described below.

### Pool of steps for image and signal processing

The algorithms presented in the related works have been decomposed into their main processing modules relative only to the automatic envelope tracing process. Hereafter, they are briefly described, following the phases reported in Fig 3. As some were used in more than one study, they are not generally associated with a specific work. We report the parameterization chosen by the original authors, when available.

**Image preprocessing phase.** This phase is performed to remove noise and emphasize aspects of interest in the image. From the literature review presented above, we know that this phase can involve up to two sub-phases, namely image filtering and contrast enhancement.

*Image filtering*: *Gaussian filter*. Image filtering techniques are traditionally associated with image enhancement, whereby a low-pass filter is generally adopted to remove high-frequency noise while preserving the frequency components at the lower end of the spectrum [24]. A Gaussian low-pass filter is a smoothing filter that produces a particular Gaussian shape of the blurred kernel whose radius is related to the standard deviation $\sigma$ of the distribution. In Refs. [18, 19, 22], the authors used a Gaussian low-pass filter with $\sigma = 1.5$, which had been determined empirically, based on a trade-off between noise suppression and blurring of the image data. As the present work does not address the fine tuning of parameters associated with the

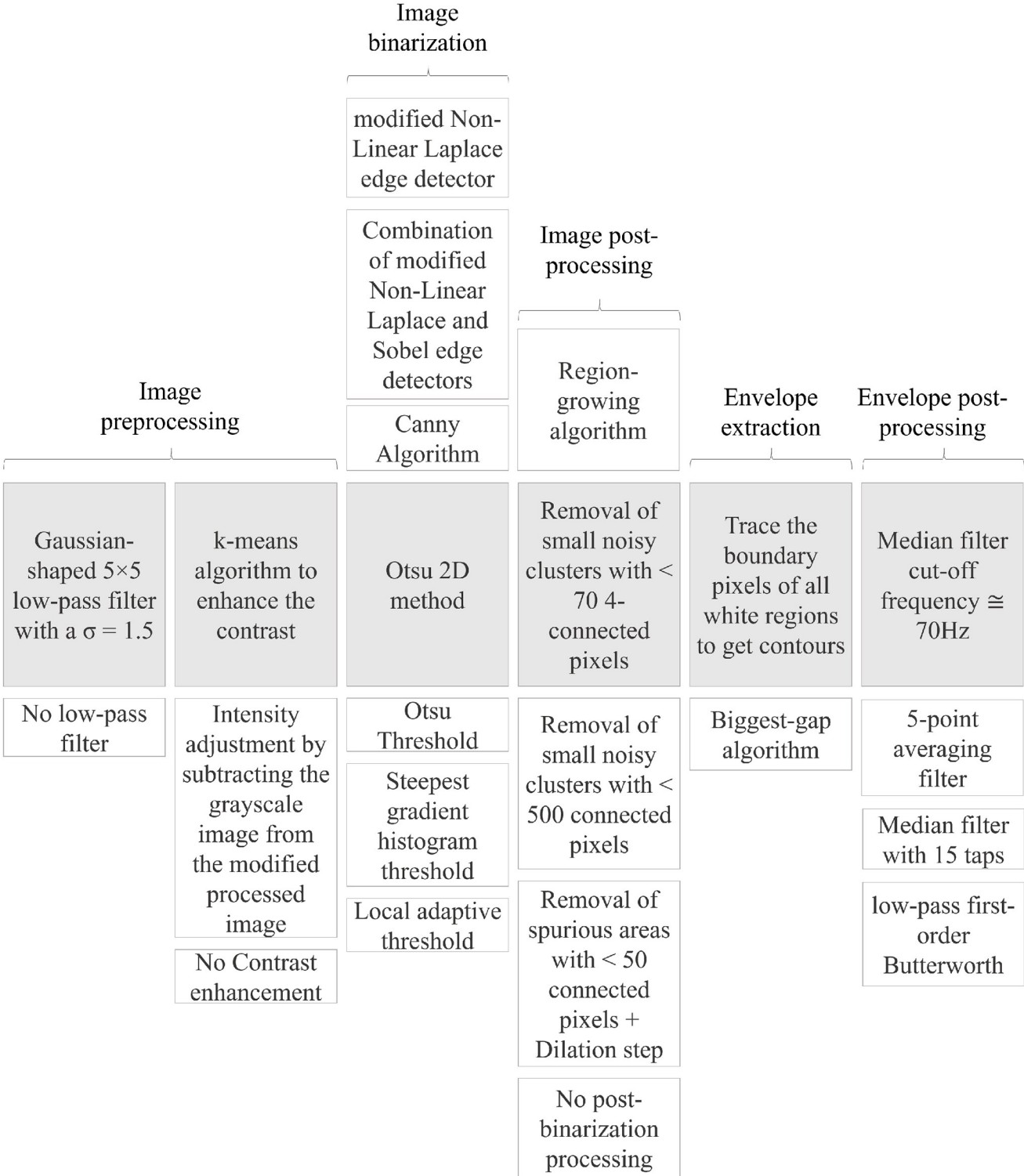

**Fig 3. Envelope tracing workflow.** Columns show the different processing steps in the workflow for tracing the Doppler velocity envelope. The shaded boxes in the center row indicates the chosen main work chain and the remaining indicate the available options from the state-of-the-art algorithms evaluated in this study.

different processing steps reported in the literature, the value of $\sigma$ adopted by the original authors is considered in our comparative assessment.

Some algorithms for Doppler envelope tracing do not include any low-pass filtering at the beginning of the processing chain, relying either on the original image quality or on the effectiveness of the downstream processing steps.

*Contrast enhancement*: *K-means*. The authors of [19] achieved both the removal of background noise and stretching of the signal contrast by the use of two thresholds, whose values were chosen based on the range in which the signal needed to be enhanced. These two thresholds were defined using a k-means algorithm that clustered pixels by their intensity into three classes, i.e., background pixels (class 1), weak-signal pixels (class 2) and strong-signal pixels (class 3). Using the values of the cluster centers, low and high thresholds can be identified. The $Th_{low}$ value s determined to be the point at which the histogram value drops by $\frac{1}{\sqrt{2}}$ its value at the class-1-cluster centroid $\mu_1$. The other threshold, $Th_{high}$, is similarly selected using the value of the class-3-cluster centroid $\mu_3$. Using both thresholds, contrast stretching is implemented as follows:

$$g(s) = \begin{cases} 0 & s < Th_{low} \\ \frac{1}{Th_{high} - Th_{low}} s - \frac{Th_{low}}{Th_{high} - Th_{low}} & Th_{low} \leq s \leq Th_{high} \\ 1 & s > Th_{high} \end{cases} \tag{1}$$

where $s$ is the pixel brightness.

*Contrast enhancement*: *Intensity adjustment*. An image must have sufficient brightness and contrast for easy viewing. Brightness refers to the overall lightness or darkness of an image, whereas contrast refers to the difference in brightness between objects or regions in the image. Adjustment of the contrast and brightness enables visualization of the features in an image [25]. The authors in [22] performed contrast adjustment by subtracting the original grayscale image from the image obtained by morphological operations using disk approximations.

We note that some algorithms used for Doppler velocity tracing have no such adjustment capability.

**Image binarization phase.** To trace the velocity profile from a Doppler image, the image is usually binarized. In general, an image can be binarized using an edge detector or a threshold technique. Specifically, segmentation is used to subdivide an image into its constituent regions or objects. Most segmentation algorithms are based on one or two basic properties related to intensity values: discontinuity and similarity. Regarding discontinuity, the idea is to partition an image based on abrupt changes in intensity, such as edges. When considering similarity, the idea is to partition an image into uniform regions according to predefined criteria, such as color, gray level, etc. Thresholding, region growing, region splitting and merging are examples of these approaches [25].

*Non-linear Laplace edge detector*. A modified version of the NLLAP edge detector [26] was adopted in [18] whereas, in [19], a combination of the NLLAP and Sobel edge detectors was proposed. The NLLAP edge detector exhibits excellent performance even at very low signal-to-noise ratios. Its algorithm, derived from the original NLLAP detector, can be described as follows:

$$NLLAP(x, y) = grandmax(x, y) + grandmin(x, y) \tag{2}$$

where:

$$grandmax(x, y) = \max_{i \in N_4}[(I(x_i, y_i)] - I(x, y) \tag{3}$$

$$grandmin(x, y) = \min_{i \in N_4}[(I(x_i, y_i)] - I(x, y) \tag{4}$$

for every pixel in the input image. The pair $(x, y)$ represents the set of coordinates of the center pixel, and $N_4$ is the set of all four neighbors connected to the center pixel. The function $I(x_i, y_i)$ indicates the intensity (brightness) of the underlying pixeland *grandmax* and *grandmin* represent the highest and lowest gradients, respectively, around the pixel located at coordinates $(x, y)$.

We note that this algorithm is adaptive by nature. At each kernel position, the neighbors delivering the highest and lowest gradients are chosen, neighbors the other neighbours do not contribute to the result for that point. The result obtained from Eq 2 consists of regions with positive and negative values, which may be separated by a band of zeros. Each pixel with a zero value is assigned to its nearest region, either positive or negative. Edges are then positioned at the locations of the zero-crossings. The strength $i_e$ of an edge is given by:

$$i_e = min[grandmax(x, y) - grandmin(x, y)] \tag{5}$$

The edge detector delivers a grayscale image with intensities corresponding to the edge strength. However, a binary image is needed to trace the Doppler envelope. Weak edges are often spurious, due to noise, which varies significantly across Doppler images. This means that the choice of an appropriate threshold for tracing the velocity envelope is directly related to the image content and must be determined adaptively. This is achieved by taking several small areas of the original image near the horizontal axis and evaluating their means and standard deviations. The sample with the lowest mean value is taken to be representative of the background and its mean ($m_b$) and standard deviation ($\sigma_b$) values are used to select the threshold value $t$ using Eq 6:

$$t = \begin{cases} l_1 & \text{if } m_b < m_{th} \text{ or } \sigma_b > \sigma_{th} \\ l_2 & \text{otherwise} \end{cases} \tag{6}$$

where $l_1$, $l_2$, $m_{th}$, and $\sigma_{th}$ are empirically determined beforehand.

*Combining NLLAP and Sobel edge detectors*. The work reported in [19] used a combination of a Sobel operator [27] and an NLLAP edge detector [26]. The Sobel edge detector [27] is a gradient-based method that uses first-order derivatives, originally approximated using $3 \times 3$ kernels [28]. As no peculiar kernel size is mentioned in [19], a size equal to $3 \times 3$ was adopted, based on the original definition of the algorithm [28]. The results can be combined to determine the absolute magnitude of the gradient, which represents output edges with larger values.

*Canny edge detector*. In Ref. [22], the authors applied Canny edge detection to binarize images. After performing noise reduction using a Gaussian filter, the Canny algorithm [29] uses four filters to identify the horizontal, vertical and oblique edges (in given ranges), which are obtained by the first derivatives of a Gaussian. Then, edge points are detected as local maxima of the derivatives (in a given direction). To trace the contours, a hysteresis-based process is controlled by two thresholds to limit fragmentation of the reconstructed edges.

*Threshold-based binarisation by Otsu threshold*. Because of its simplicity and low computational cost, image thresholding holds a central position in image segmentation applications [24]. Several threshold techniques have been presented in the literature with respect to Doppler velocity-envelope tracing, which we used in this work. The first technique is the Otsu threshold

[30], which is based on the assumption that the histogram of a given image is bimodal, so that two classes of pixels (background and content of interest) can be identified. To this end, the optimum threshold is determined for separating the two classes, with minimal intra-class variance. This threshold was employed by Ref. [20].

The Doppler spectrum covers a large portion of the grayscale range. To effectively separate information of interest, in Refs. [16, 17], the image is binarized using a global threshold obtained from a gray-level-median histogram based on the 2D Otsu method. In this method, a 2D histogram is built, but instead of using the gray level and average gray value of the neighborhood of image pixels, the gray level and median of the neighborhood of image pixels are used, which effectively separates noise pixels from object and background regions [31].

*Threshold-based binarization by steepest gradient histogram.* In Ref. [23] the optimum value is empirically selected for contrast thresholding using the steepest gradient method. To separate the foreground from the background, 80% was suggested to be a good threshold for the x-intercept of the steepest tangential line to the gradient of the image histogram [32]. However, in the original work, the authors considered the optimum threshold to be the value at which the gradient of the image histogram reaches its peak, which indicates the background pixel values in a grayscale Doppler image.

*Threshold-based binarization by local adaptive threshold.* To automatically extract the Doppler velocity profile from Doppler images, the authors in [21] applied a local adaptive thresholding algorithm. Adaptive thresholding takes into account the spatial variations in illumination. To account for these variations, a different threshold is computed for each pixel in the image. In their study, the authors first divided the region of interest (ROI) into three partially overlapping and rectangular regions of equal size. Then, for each ROI, an inverse normalized cumulative intensity histogram was computed. The threshold for a given ROI was estimated to be the 25$^{th}$ percentile of the pixel intensities. In overlapping areas, they used the average thresholds of the two adjacent regions.

**Image post-processing phase.** *Removal of spurious areas by connected pixel analysis.* Mathematical morphological operations are commonly used as image processing tools for extracting image components that are useful for representing and describing a regional shape. Image enhancement using morphological approaches has been widely addressed in previous experimental research on Doppler velocity-envelope tracing, primarily to reduce residual noise [33]. The two most common types of additive noises are Gaussian and salt-and-pepper noise [33]. Although noise is ordinarily removed by spatial or frequency-domain filters, morphological processing may be more frequently adopted to reduce noise in binary images. In this processing, a dilation step is applied to fill small holes in bright regions to yield a clean velocity signal. To filter out small noisy clusters, connected areas with fewer than a predefined number of pixels are removed; this number was empirically selected to be 500 pixels in [23], 50 in [20], and 70 in [16].

*Region growing.* Region growing [34], a simple region-based image segmentation method, was used in the post-processing image phase by the study reported in [18]. This segmentation approach examines the neighboring pixels of initial seed points and determines whether these pixel neighbors should be included in the region. Starting from a seed-point, the flood-fill algorithm finds all the connected pixels of the same value and deletes small spurious areas from the image.

We note that some algorithms used for Doppler velocity tracing include no post-processing step.

**Envelope extraction phase.** The regional outline of an image differs from the background, which should provide a good estimate of the Doppler velocity profile, and can be extracted in two lines representing the upper and lower profiles. In the literature, two main

envelope extraction methods have been presented, the biggest-gap method and the white-region contour method, which are described below.

*Biggest-gap method*. The biggest-gap method [19] defines a column vector that moves across the image from left to right. Within this column vector, the sub-vector with the highest number of consecutive black pixels is identified. Consecutive black pixels comprise a gap. The biggest-gap algorithm weights gap values according to their vertical positions, so that a gap in the central area of the search space is more significant than those in the upper and lower parts. The lowest [highest] pixel of the largest gap, considering all the position-dependent weighted gaps, is labeled as part of the upper [lower] envelope function.

*White-region contour method*. This method, used by the authors in [16, 23], traces the boundary pixels of all white regions to obtain contours as follows:

$$G_u(x) = \operatorname*{argmax}_y\{I(x, y) = 1\} - y_b \tag{7}$$

$$G_l(x) = \operatorname*{argmin}_y\{I(x, y) = 1\} - y_b \tag{8}$$

where $y_b$ is the baseline position, i.e., the horizontal axis line that divides the image into two parts. Depending on the direction of the blood flow away from or toward the transducer, one part is characterized by positive waves and the other by negative waves.

**Envelope post-processing phase.** Lastly, to remove the effect of spiking artefacts or to smooth the velocity profile signal, a post-processing step can be applied to the extracted envelope. In this study, we compared four different filters, a median filter of order 15, a five-point averaging filter, a median filter with a window size of two samples and a cut-off frequency of 71 Hz, and a first-order Butterworth filter with a cut-off frequency of 70 Hz.

*Median filter*. A basic processing step is the application of a temporal median filter with a variable window size. The median filter evaluates the median value within a sliding window of a given size, and provides a smoothing effect that is less sensitive to outlier values than the moving average filter.

In [21], the extracted envelope is filtered using a median filter with order 15, whereas in [20] a median filter with a window size of three pixels is used to remove the effect of spiking artefacts on the extracted envelopes. In the latter case, when considering a sampling frequency of 500 Hz, the obtained cut-off frequency was approximately 78 Hz.

*Butterworth filter*. In [23], the authors adopted a Butterworth low-pass filter to suppress high-amplitude outliers and artefacts in the velocity profile. Any frequency ten times higher than the fundamental frequency of the heart motion was filtered out. This ratio was selected empirically as a trade-off between noise removal and waveform accuracy. The low-pass filter is also a crucial tool for isolating individual cardiac cycles.

*Moving average filter*. To suppress the effect of remaining noise, Ref. [18] applied a five-point averaging filter to the envelope curve. This simple filter smooths the extracted envelope much like other kinds of low-pass filters, but with a simpler implementation.

## Performance assessment

To enable evaluation of the role played by the processing steps of each option described above with respect to PWD envelope tracing, we identified a prototypical processing chain (highlighted in gray in Fig 3), which we labeled *the main work chain* (MC).

Starting from the MC, step by step from first to last, we evaluated each processing option by introducing it into the MC (replacing only the homologous step in the MC). As the MC comprises six steps, the number of chains composed of five steps belonging to the MC and one step

| MC | Gaussian-shaped 5x5 low-pass filter with a σ = 1.5 | k-means algorithm to enhance the contrast | Otsu 2D method | Removal of small noisy clusters with < 70 4-connected pixels | Trace the boundary pixels of all white regions to get contours | Median filter cut-off frequency ≅ 70Hz |
|---|---|---|---|---|---|---|
| 1 | No low-pass filter | k-means algorithm to enhance the contrast | Otsu 2D method | Removal of small noisy clusters with < 70 4-connected pixels | Trace the boundary pixels of all white regions to get contours | Median filter cut-off frequency ≅ 70Hz |
| 2 | Gaussian-shaped 5x5 low-pass filter with a σ = 1.5 | Intensity adjustment by subtracting the grayscale image from the modified processed image | Otsu 2D method | Removal of small noisy clusters with < 70 4-connected pixels | Trace the boundary pixels of all white regions to get contours | Median filter cut-off frequency ≅ 70Hz |
| ⋮ | ⋮ | ⋮ | ⋮ | ⋮ | ⋮ | ⋮ |
| 17 | Gaussian-shaped 5x5 low-pass filter with a σ = 1.5 | k-means algorithm to enhance the contrast | Otsu 2D method | Removal of small noisy clusters with < 70 4-connected pixels | Trace the boundary pixels of all white regions to get contours | low-pass first-order Butterworth |

**Fig 4. Workflow of the classification method.** Example of the procedure used to evaluate the importance of each single processing technique. Specifically, the processing chains created by MC, $1^{st}$, $2^{nd}$, and $17^{th}$ are shown.

from the available pool of steps will be 23 − 6 = 17. A workflow example is shown in Fig 4 where, from top to bottom, we can see the MC, followed by the first, second, and last ($17^{th}$) chain created by this procedure. Table 2 lists the different processing chains constructed in this way. This numbering system is used to present of the results of our analysis.

A second assessment was aimed at the identification of a near-optimal work chain, without an exhaustive exploration of the design space. In this case, we tested the different options for the first processing step by the MC, selected that leading to the best performance and used it to modify the MC. With this change to the MC, we tested the different options for the second processing step, changing the MC to include that leading to the best performance. This step-wise process was repeated six times. Although this solution does not guarantee that the best

**Table 2. Processing chains for assessment of individual steps.** The numbering and description of the different processing chains created to assess the importance of individual steps in a prototypical processing chain.

| # | Description |
|---|---|
| MC | A Gaussian-shaped low-pass filter with $\sigma = 1.5$, a k-means algorithm to enhance the contrast, application of the Otsu 2D method, removal of small noisy clusters with 70 four-connected pixels, tracing of boundary pixels of all white regions to obtain contours, and a temporal median filter with a window size of three pixels. |
| 1 | MC with no filter application for the first step of the preprocessing phase |
| 2 | MC that performs intensity adjustment by subtracting the grayscale and modified images in the second step of the preprocessing phase |
| 3 | MC that uses no operation for the second step of the preprocessing phase |
| 4 | MC that uses the Canny algorithm for the image binarization phase |
| 5 | MC that uses the NLLAP edge detector for the image binarization phase |
| 6 | MC that uses a combination of the NLLAP and Sobel edge detectors for the image binarization phase |
| 7 | MC that uses an adaptive threshold for the image binarization phase |
| 8 | MC that uses the steepest gradient histogram threshold for the image binarization phase |
| 9 | MC that uses the Otsu threshold for the image binarization phase |
| 10 | MC that uses the region-growing algorithm in the image post-processing phase |
| 11 | MC that removes spurious areas by performing a morphological operation (MO) with a maximum of 500-connected pixels in the image post-processing phase |
| 12 | MC that removes spurious areas by performing a morphological operation (MO) with a maximum of 50-connected pixels in the image post-processing phase |
| 13 | MC without any image post-processing phase |
| 14 | MC that uses the biggest-gap algorithm for envelope extraction |
| 15 | MC that uses a median filter of five-point length for envelope post-processing |
| 16 | MC that uses a median filter of 15-point length for envelope post-processing |
| 17 | MC that uses a low-pass first-order Butterworth filter for envelope post-processing |

processing chain in the whole design space will be obtained, it enables the identification of a near-optimal solution (hereafter referred to as the sub-optimal chain, SOC).

Lastly, we compared the different state-of-the-art chains (listed and described in Table 1), the SOC, and a randomly built work chain. The latter was included to verify whether or not the robustness of the algorithms was such that the actual composition of the processing chain was of little importance, provided that the processing elements are taken from the pool, as described in this section.

The aim of our study is not the identification of a solution leading to the best envelope tracing algorithm, but rather the identification of the traced envelope that maximizes the performance of a generic classifier trained to distinguish between fetal heartbeat with complete and clinically meaningful PWD segments and those with incomplete or malformed segments. For this reason, the assessment focuses on the accuracy of a chosen classifier, whose architecture is presented in the next section. The minimum value for the accuracy considered as acceptable in this work was 90%.

**Classifier model.**   In this work, we used a simple multi-layer perceptron ANN to identify the portions of the PWD envelope in which there is a recognizable complete and clinically well-formed fetal cardiac cycle. ANNs are characterized by a variable number of hidden layers and a variable number of nodes comprising the input, hidden, and output layers. Using the scaled conjugate gradient back-propagation algorithm, we trained an ANN characterized by 264 input nodes, according to the number of extracted features, 10 hidden nodes, and two output nodes, which equal the number of classes that we wished to identify.

In particular, the input fed to the ANN is as follows:

1. the upper $G_u(x)$ and the lower $G_l(x)$ envelopes in a 128-point window. These two envelopes have been normalized to between 0 and 1 and between -1 and 0, respectively, without any resampling. Based on the window size, a total of 256 inputs are associated with the traced PWD envelopes;

2. four features associated with the area under the curve of the two envelopes in the first and second halves of the window ($A_i$);

3. four pixel-based features ($P_i$), computed as the mean values of the pixels included in the four areas described in the previous point.

Overall, 264 input features for every sample are then presented to the ANN to be classified.

By considering the envelope sampling rate and the duration of the mean cardiac cycle among the targeted gestational weeks, the chosen window length can capture a whole fetal heartbeat. Moreover, considering only healthy fetuses, a 1:1 atrio-ventricular conduction can be assumed so that the atrial activity can be enclosed in one half of the window and the ventricular activity in the other half.

Fig 5 shows the structure of the ANN used and the feature extraction step. The number of neurons in the hidden layer and the number of hidden layers were empirically chosen to obtain acceptable performance on the available dataset without incurring overfitting problems [17, 35].

**Performance metrics and statistical analysis.** To evaluate the impact of the different steps on the whole processing chain, we determined the accuracy (Acc) of the classification as follows:

$$Acc = \frac{TP + TN}{P + N} \cdot 100 \tag{9}$$

where TP is the number of true positive detections, TN is the number of true negative detections, P is the number of positive samples and N is the number of negative samples.

We adopted a leave-one-subject-out cross-validation scheme corresponding to a 25-fold cross-validation, but without any randomness in the selection of the samples from the test set. This approach was considered to be fairer than a normal k-fold stratified cross-validation as the leave-one-subject-out approach does not train the classifier on any heartbeat coming from a given subject, which can happen when this type of system is integrated into an ultrasound device. Using this validation scheme, we obtained 25 Acc values. To observe the symmetry of the distributions, we adopted box-and-whiskers plots, as shown in Figs 8 and 9. In these plots, the median value is highlighted inside a box ranging from the 25th to 75th percentiles and the whiskers range from the minimum to maximum number of samples that cannot be considered to be outliers (represented by red crosses).

Statistical analyses were performed to support the interpretation of the obtained results. First, we used the Lilliefors test to determine the normality of the distributions. If a result did not satisfy the assumption of a normal distribution, a non-parametric statistical test was applied. Specifically, we used the Kruskal–Wallis test for comparisons involving more than two distributions to verify that they had all come from the same distribution. Otherwise, we used the Wilcoxon signed rank test to compare just two distributions to verify the statistical difference between the two. When the results of the Lilliefors test satisfied the assumption of a normal distribution, we then applied the Student's t-test for paired data. In all the statistical tests, we considered $p < 0.05$ to indicate statistical significance.

All data processing was performed using MATLAB v2018b (MathWorks Inc., MA, USA).

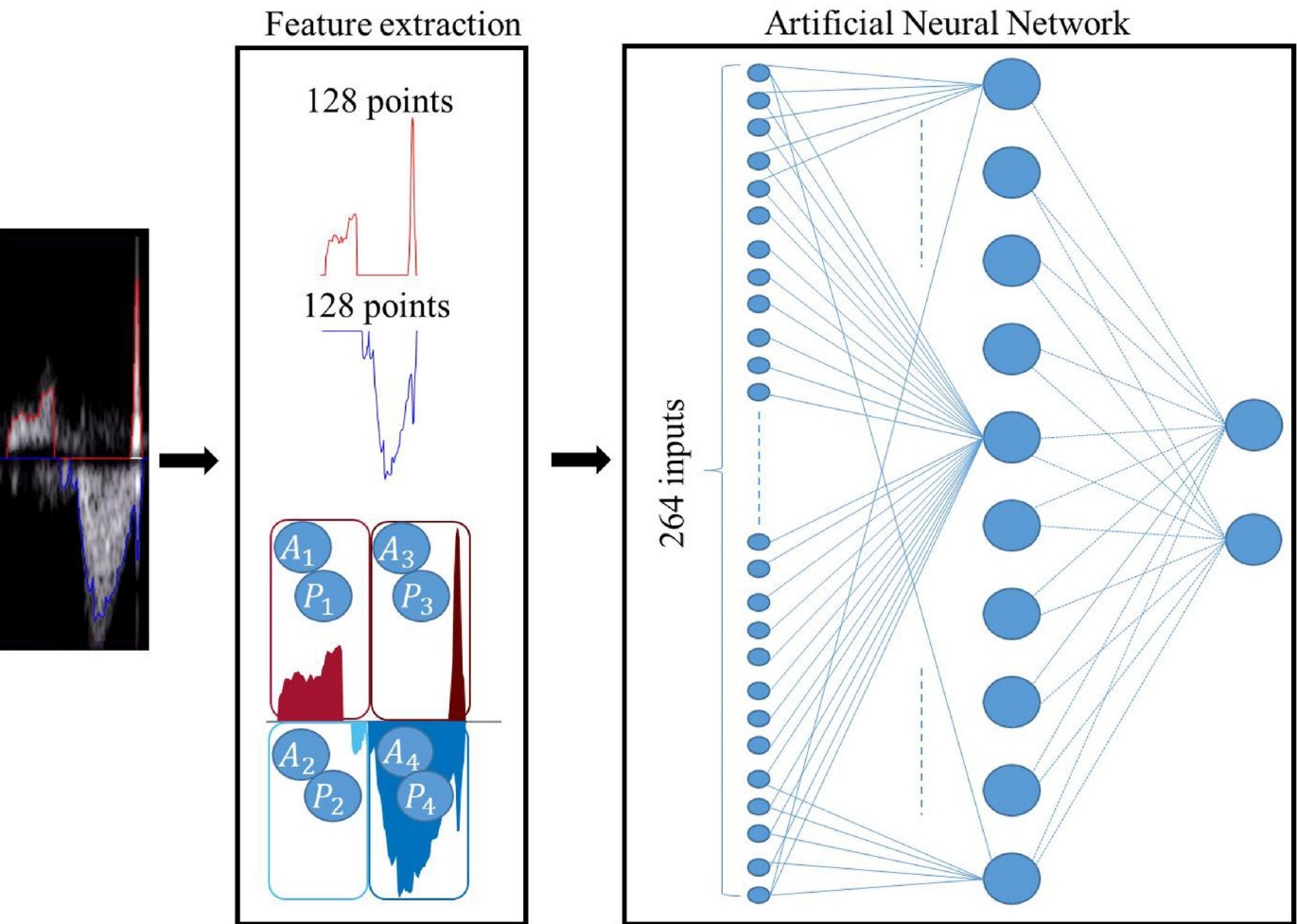

**Fig 5. Classifier model.** From left to right, the features are extracted from a portion of the PWD signal (in this case that representing a meaningful beat). The features include 128 samples from the upper envelope (red line), 128 samples from the lower envelope (blue line), four area features ($A_i$), and four pixel-based features ($P_i$). The ANN is fed by this 264-element feature vector, and consists of one hidden layer with 10 hidden neurons and two output nodes for binary classification.

**Test dataset of fetal PWD.**    A dataset of fetal cardiac PWD signals was collected at the Division of Pediatric Cardiology, San Michele Hospital, Cagliari, Italy. The study was approved by the Independent Ethics Committee of the Cagliari University Hospital (AOU Cagliari) and performed following the principles outlined in the Helsinki Declaration of 1975, as revised in 2000. Each volunteer signed a form acknowledging their informed consent to the research protocol. All images and data were anonymized prior to analysis.

The PWD signals were recorded using a Philips iE33 ultrasound machine (Philips, The Netherlands). We set the sweep speed to 75 mm/s and the machine settings (e.g., gain, axis scaling, and baseline) were not changed during the acquisition process (ranging from 6.4 s to 119.8 s). All the frames were captured using the video frame grabber USB3HDCAP USB3.0 (by StarTech, Ontario, Canada), which was connected to the DVI output of the ultrasound device. To ensure that no frame was dropped, the frames were collected at a rate of 60 fps, which is higher than both the screen refresh rate and the DVI output (30 fps).

Based on best clinical practice, we chose the five-chamber apical window. Due to the variable position of the fetus in the maternal uterus, the mitral blood inflow can be moving either towards

or away from the ultrasound transducer. Therefore, the E/A-V pattern can have a positive balance (positive E/A wave, negative V wave) or a negative balance (negative E/A wave, positive V wave). We converted the whole video into a single wide image [17] and, to work only with images exhibiting a positive balance, we flipped the negatively balanced images upside-down.

The data were collected from the fetal echocardiographic examination of 25 low-risk pregnant volunteers at gestational weeks ranging from the 21$^{st}$ to the 27$^{th}$, from which we obtained a total of 43 PWD traces. An expert pediatric cardiologist labeled all the complete and measurable fetal cardiac cycles using a custom MATLAB graphical interface, as described in [15]. This MATLAB application enables the cardiologist to scroll through the traces and label fetal heartbeats frame-by-frame. The graphical interface drew two rectangular outlines on the PWD image, one enclosing the atrial activity and the other the ventricular activity. The window size was set to be 128 samples wide, as previously explained. The cardiologist could label a selected window as "complete and measurable" or continue to scroll over the trace.

Overall, 174,319 PWD windows were labeled. For each manually labeled complete and meaningful fetal cardiac cycle, the tool also labelled other windows, as many as 15, in the same way before and after the one manually labeled by the cardiologist. Overall, we obtained 87,736 windows representing meaningful cycles. The other 86,583 windows representing incomplete or malformed fetal cardiac cycles were randomly selected by the application. This approach yielded to a balanced dataset.

Fig 6 shows an example of two fetal heartbeats: one that is complete and measurable by the cardiologist and the other that shows an incomplete fetal cycle. The complete heartbeat allows for the clear identification of the atrial and ventricular activity of the fetal heart, whereas the other does not. The atrial activity must also contain distinguishable A and E waves that can be labelled as measurable by the cardiologist.

## Results

### Assessment of individual steps

Fig 7 shows the output image and traced envelopes obtained by the MC and the other 17 processing chains constructed for this first assessment. For the sake of clarity, the processed images shown were obtained from the same exemplary PWD segment shown in Fig 2. It is clear by observation alone that some chains realized better tracing than others, even apart from the analysis of the classifier output.

Fig 8 shows the results of this first analysis, in terms of classifier accuracy, grouped by processing step, which enables a direct comparison with the MC. The distribution of the accuracy values related to the third and fifth steps in Fig 8 clearly reveal that two main steps have a major influence on classification: image binarization and envelope extraction. Table 3 shows the results of the statistical analysis of this first assessment. The results of the Kruskal-Wallis test confirm a significant difference between those two steps ($p < 0.001$). At the same time, the results of the pairwise Wilcoxon tests, which was applied to each processing step on the MC and other processing chains that differed only in the use of another option in that step, reveal significant differences for all combinations in the phases mentioned above and between chains 2, 15, and 16 and the MC.

Table 4 presents a comparative analysis of all 18 processing chains in terms of the median and first and third quartiles.

### Identification of a near-optimal processing chain

Table 5 presents the results regarding the identification of an alternative, near-optimal, processing chain, which was created following the approach described in the previous section.

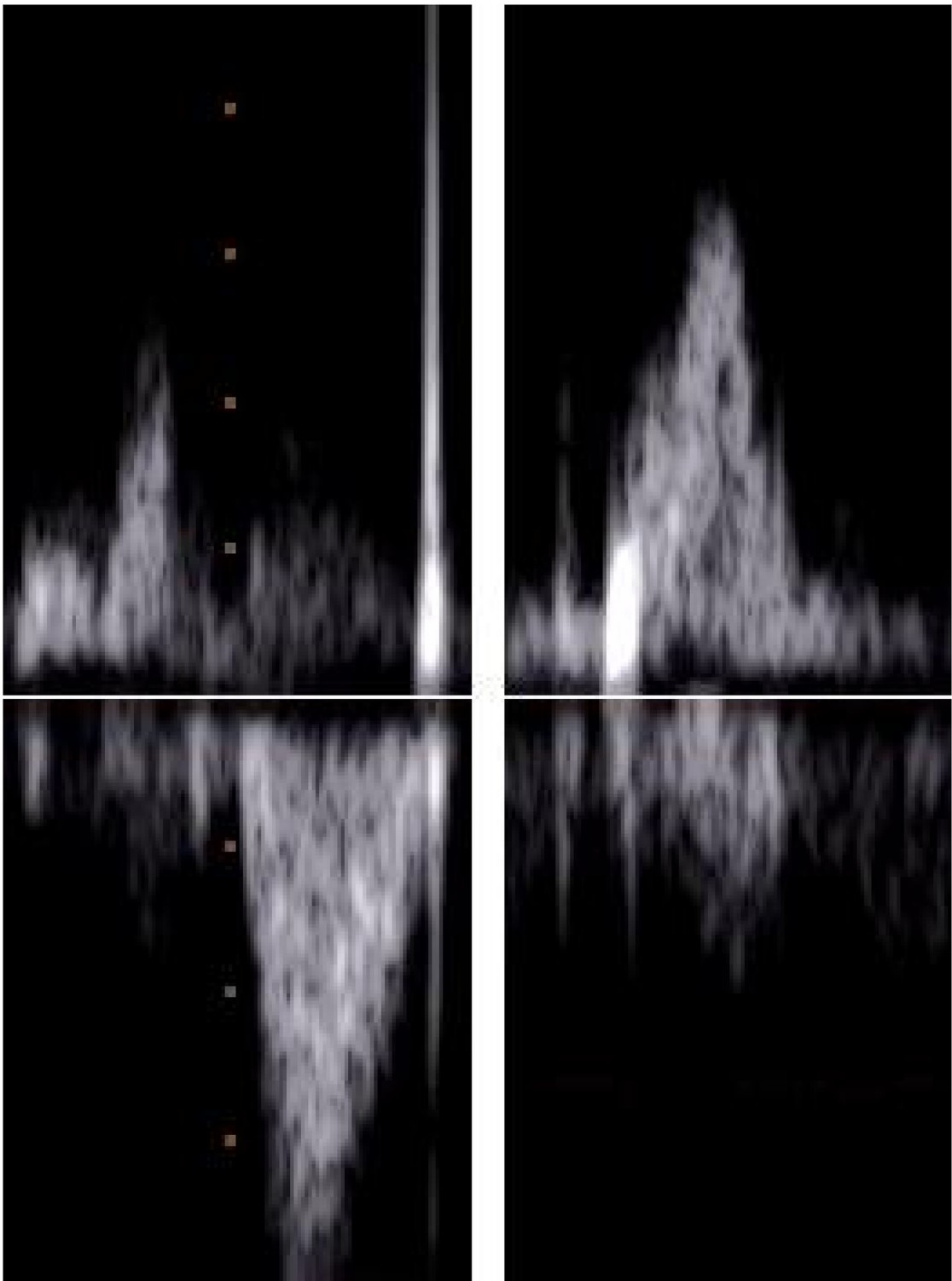

**Fig 6. Examples of complete and incomplete fetal cardiac cycles.** A PWD segment labeled as complete and measurable by the cardiologist, with clear E/A and V patterns (left), and a PWD segment labeled as incomplete/malformed (right).

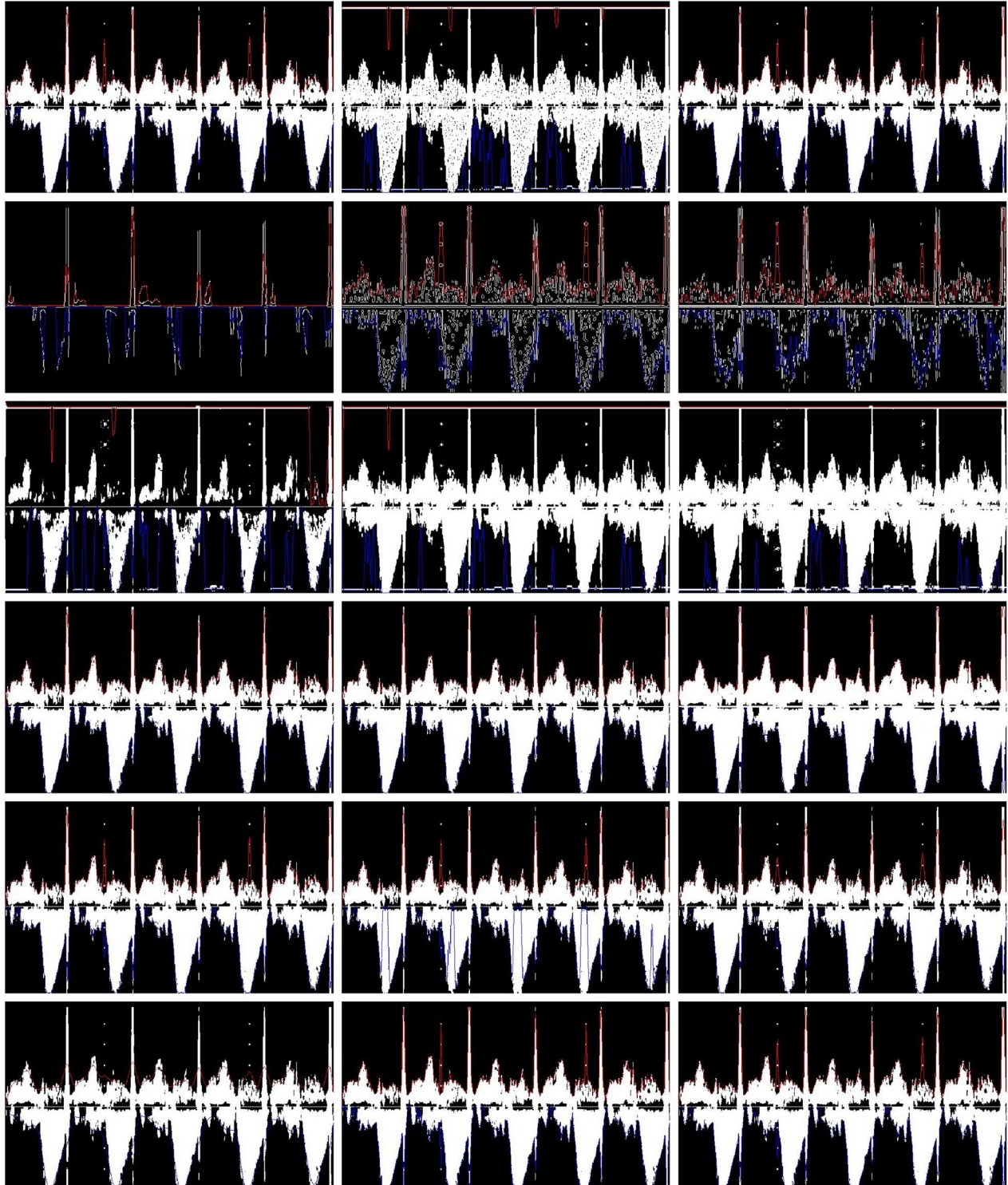

**Fig 7. Example of output images.** Each image presents envelopes (upper envelopes (red) and lower envelopes (blue)) and a binarized image, as obtained by the different signal processing chains listed in Table 2, for the same five fetal beats. From left to right, and top to bottom, the processing chains are identified by numbers 1 to 17 in Table 2. The last image was obtained using the MC.

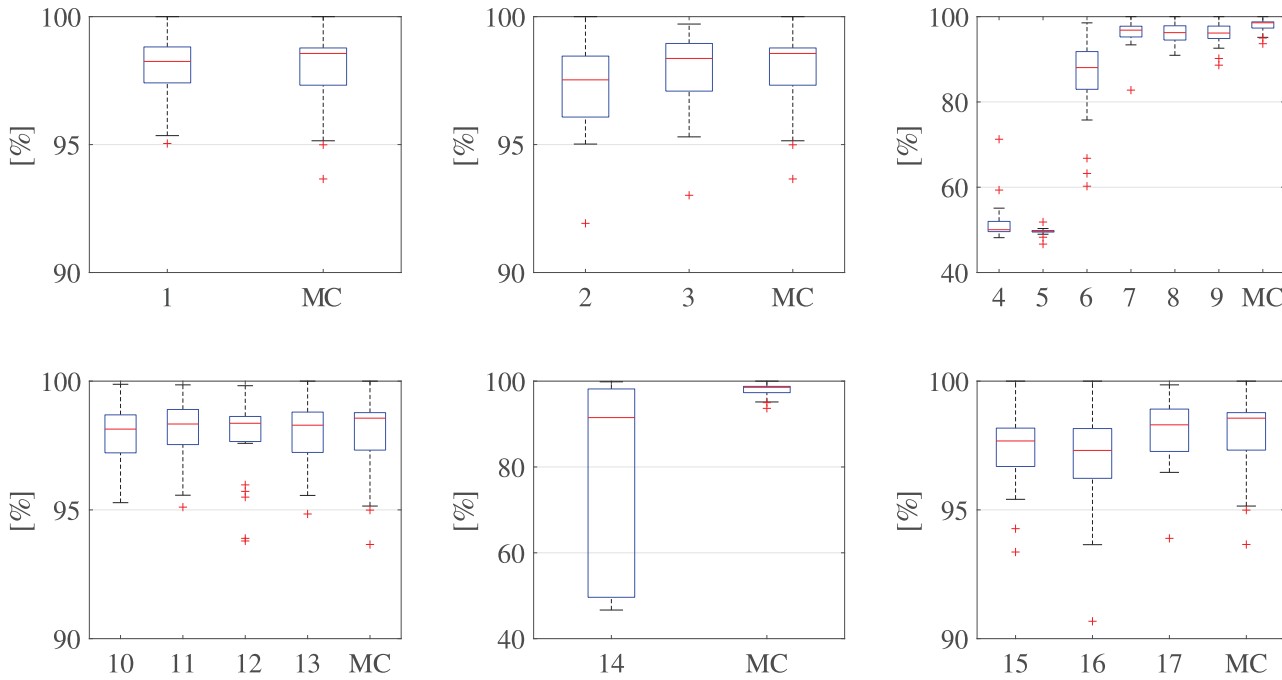

**Fig 8. Assessment of individual processing steps.** Accuracies (percentages) of the six steps (from left to right, top to bottom) identified in Fig 3. For the numbering used to identify these chains, refer to Table 2.

The last row of this table lists the p-values obtained from the Kruskal–Wallis test for all the options available for that step to determine whether an observed difference introduced by a given choice was statistically significant. We identified the most accurate processing method for a single step by selecting the one that achieved the highest accuracy (based on the mean of the 25-fold validation).

Based on this approach, the resulting SOC was identified. Step 1: application of Gaussian low-pass filter; Step 2: k-means clustering contrast adjustment; Step 3: binarization using 2D Otsu thresholding); Step 4: removal of spurious areas (less than 500 pixels); Step 5: tracing of the boundary pixels of all white regions to obtain contour; and Step 6: smoothing of the envelope by the Butterworth filter.

## Comparison of different processing chains

The last analysis we performed was a comparison of the accuracies of the processing chains presented in the literature (see Table 1) with the SOC and a randomly-composed chain. Fig 9 shows the results of this comparison.

Based on their performances, the Kruskal–Wallis test clearly indicates that the distributions of the tested chains are statistically different ($p < 0.001$). Moreover, we we used the Wilcoxon

**Table 3. Statistical analysis of individual step assessments.** For each step identified in the first row of table (Fig 3), the second row shows the Kruskal–Wallis test result, and the third row uses an * symbol to indicate any statistically significant difference in the results of the pairwise Wilcoxon signed rank tests between the MC and the other processing chains that differ only in the use of another option in that step.

| 1st | 2nd | 3rd | 4th | 5th | 6th |
|---|---|---|---|---|---|
| $p > 0.05$ | $p > 0.05$ | $p < 0.001$ | $p > 0.05$ | $p < 0.001$ | $p > 0.05$ |
| 1 vs. MC | 2* 3 vs. MC | 4* 5* 6* 7* 8* 9* vs. MC | 10 11 12 13 vs. MC | 14* vs. MC | 15* 16* 17 vs. MC |

**Table 4. Details of individual step assessment results.** Q1 refers to the first quartile, Q2 to the median and Q3 to third quartile. The results are expressed as percentages.

| | 1 | 2 | 3 | 4 | 5 | 6 | 7 | 8 | 9 | 10 | 11 | 12 | 13 | 14 | 15 | 16 | 17 | MC |
|---|---|---|---|---|---|---|---|---|---|---|---|---|---|---|---|---|---|---|
| **Q1** | 97.4 | 96.1 | 97.1 | 49.6 | 49.5 | 83.0 | 95.2 | 94.5 | 94.9 | 97.2 | 97.5 | 97.7 | 97.2 | 49.6 | 96.7 | 96.2 | 97.3 | 97.3 |
| **Q2** | 98.3 | 97.5 | 98.4 | 50.0 | 49.7 | 88.1 | 96.8 | 96.3 | 96.2 | 98.1 | 98.3 | 98.4 | 98.3 | 91.1 | 97.7 | 97.1 | 98.3 | 98.6 |
| **Q3** | 98.8 | 98.5 | 99.0 | 52.0 | 49.8 | 91.8 | 97.8 | 97.9 | 97.8 | 98.7 | 98.9 | 98.6 | 98.8 | 98.2 | 98.2 | 98.2 | 98.9 | 98.8 |

**Table 5. Near-optimal chain identification results.** For each processing step identified in Fig 3, the accuracy is reported (as a percentage). The number in the brackets indicates the processing chain in Table 2 from which that processing option was taken. The results in bold indicate the option selected for use in successive tests.

| Step 1 | Step 2 | Step 3 | Step 4 | Step 5 | Step 6 |
|---|---|---|---|---|---|
| 98.33 (1) | 98.12 (2) | 51.72 (4) | 97.83 (10) | 49.79 (14) | 98.35 (15) |
| **98.36** (MC) | 96.15 (3) | 49.58 (5) | **98.06** (11) | **98.44** (MC) | 98.44 (16) |
| | **98.44** (MC) | 85.33 (6) | 97.77 (12) | | 98.05 (17) |
| | | 96.08 (7) | 97.95 (13) | | **98.45** (MC) |
| | | 95.98 (8) | 97.93 (MC) | | |
| | | 95.96 (9) | | | |
| | | **97.93** (MC) | | | |
| $p = 0.54$ | $p = 0.38$ | $p < 0.001$ | $p < 0.001$ | $p < 0.001$ | $p = 0.99$ |

signed rank test to make a pairwise comparison of all the distributions with that of the SOC. The results of all the comparisons, except that for the chain proposed by Sulas et al. [16], revealed statistically significant differences ($p < 0.05$). This is not surprising because the chain proposed by Sulas et al. [16] and the SOC adopted the same methods for image binarization and envelope extraction, which were identified as the most important steps in the work chain.

## Discussion

From the first analysis of the assessment of each processing option available on the processing chain, from Fig 8 and Table 3 we can conclude that, regarding image preprocessing and post-processing, performing no operation or trying to improve the image quality to emphasise the signal of interest has no significant effect on the classification outcome. This finding also applies to the post-processing of the traced envelopes, for which the performance improvement is very limited.

Regarding image binarization, we can observe that the Canny and NLLAP edge detectors are not reliable for PWD images. The combination of the NLLAP and Sobel edge detectors improved the output, compared to that of NLLAP alone, thus confirming the conclusions drawn by [19]. However, the Otsu 2D method demonstrates clear superiority ($p < 0.001$). Considering envelope extraction, our comparison of the two methods reveals the clear superiority of the white-region contour method ($p < 0.001$).

These findings are confirmed by the identification of an SOC that incorporates the best options available in those two important stages.

The results also reveal that the chains presented by Tschirren et al. [18], Greenspan et al. [19], and Zolgharni et al. [23], and the random chain obtain accuracies far below minimum expectations. Although the techniques presented by Syeda-Mahmood et al. [20] and by Kiruthika et al. [22] achieved high median accuracies, exceeding 90%, the accuracies achieved in the 25 folds in the cross-validations are widely dispersed, as can be observed from the boxplot in Fig 9. This result confirms that image binarization is a critical step and that it is better to use a thresholding technique than an edge detector for the purposes of PWD signals.

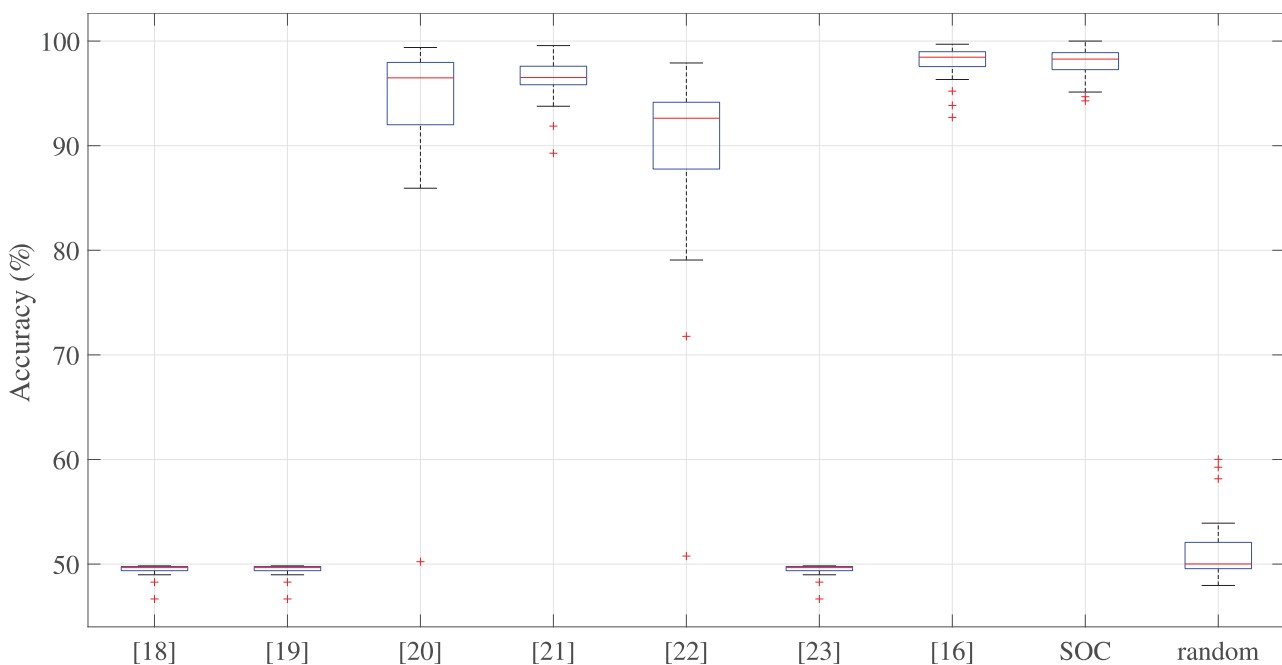

**Fig 9. Comparison of the different processing chains results.** Comparison of classification accuracies of the workflows presented in the literature (Table 1) with those of the near-optimal chain and a randomly-composed chain.

The chain proposed by Magagnin et al. [21] uses the same envelope extraction adopted by Sulas et al. [16] and the SOC, but not the same binarization technique (adaptive thresholding rather than the Otsu 2D). If we look at Fig 8, the adaptive threshold (used in chain number 7 (Table 2), whereby the MC was modified to introduce an adaptive threshold,) and the Otsu 2D (binarization algorithm used in the MC) have similar distributions. Moreover, from Table 5, we can see that the adaptive threshold (equivalent to the chain 7) achieved high classification accuracy.

From a methodological perspective, some choices made in this work require a justification. First, a fine tuning of the parameters adopted by the different processing chains and the identification of the best possible processing chain are beyond the scope of this work, which primarily focused on analysis of the importance of the different options reported in the literature for the automatic tracing of Doppler traces. In this sense, the adoption of different classifiers was not considered either. The choice of supervised classification method has been carefully studied and analyzed in previous work. Preliminary studies [15] compared three different classifier models, based on the preliminary extraction of PWD velocity envelopes. The adoption of a supervised classifier trained with selected features achieved significantly better results ($p < 0.0001$) than other feature-based approaches. The choice of the ANN has also been addressed in other work [17, 35], and its excellent performance revealed.

For the same reason, the study of deep neural network approaches was not a focus of this work, although it would be of interest to do so in future studies. In fact, while deep learning and convolutional neural networks could lead to similar or even better results, the performance improvement would be modest and would not counterbalance the significantly higher complexity of the classifier model. Moreover, these approaches would provide no insight regarding the role of the different processing steps normally used to trace the PWD envelopes.

The identification of a "universally best" processing approach could not be based on the exploration of a single dataset, recorded and annotated by a single cardiologist using a single machine with a limited number of participants. All these limitations are described below.

One limitation in this study was the adoption of a single ultrasound machine. Different ultrasound machines feature a different number of pixels per second, which lead to a different sampling frequencies of the envelopes obtained by our approach. For this reason, the size of the selected window must be adequate. The colors, brightness, and contrast of different machines could also differ and, as they are used by the ANN as features, this could lead to slightly different classifier performances. The filters are also different in different ultrasound machines. Although these aspects should be carefully considered, our study demonstrated that the first preprocessing steps that use image filters and techniques to enhance the image contrast, do not play an important role in the classification output. Therefore, it is reasonable to assume that different filters and contrasts will not have a significant impact on the extraction output.

The limitation associated with the labeling having been performed by a single cardiologist is also of little importance. No measurements were performed during the labeling procedure, merely visual inspection. From this perspective, the cardiologist's experience and habits have a negligible impact on the procedure, although they could severely impact a measurement session. This limitation is also of minor importance considering that our test dataset comprised signals from healthy fetuses only, as per the experimental protocol approved by the Ethics Committee. A more comprehensive study including a larger population with different heart conditions and at different gestational ages could improve the generalizability of the results.

Another limitation of the study is its sample size. For this pilot study, the sample size was based on a convenience sample of 25 volunteers with consideration of the pregnancy's flow, the research budget, and the time frame. The number of volunteers was determined by their availability within the time frame (about two years) allocated to the recording protocol. Very little is known about the standard deviation of the variables of interest adopted in this study, and the obtained results can be used to plan further research and perform power analyses, which was impossible for our study. We note that this type of PWD signal is quite difficult to acquire as the cardiologist must have the necessary expertise and training and there are other factors affecting the measurement, such as fetal movements and fetal heart size. Beyond the signal acquisition, which is particularly cumbersome at that gestational age, the pediatric cardiologist must label the dataset, which is a very time-consuming task.

Lastly, our assessment was performed by offline processing, as the focus was to evaluate the importance of each step in the envelope tracing process from the PWD image for the recognition of clinically complete and measurable fetal heartbeats. Therefore, the computational time effort was not considered in the evaluation of the processing chains. For such a comparison, the algorithms must also be optimized in the same way, and executed on a reference architecture. This cannot be realized by the adoption of functional MATLAB implementations. In any case, Table 6 shows the time (in seconds) to process a short Doppler image in Fig 2 containing five fetal heart cycles, executed on a Intel Core i7 with 16GB RAM. The computational effort is mostly expended during the binarization step (which changed between the 4, 5, 6, 7, 8, 9, and MC processing chains). Table 7 illustrates the same analysis for state-of-the-art chains and the

**Table 6. Processing time (seconds) for the five fetal heartbeats in Fig 2, for the 18 processing chains obtained from the MC.**

| 1 | 2 | 3 | 4 | 5 | 6 | 7 | 8 | 9 | 10 | 11 | 12 | 13 | 14 | 15 | 16 | 17 | MC |
|---|---|---|---|---|---|---|---|---|----|----|----|----|----|----|----|----|----|
| 26.32 | 30.43 | 30.63 | 0.18 | 0.44 | 0.51 | 0.14 | 0.12 | 0.10 | 32.34 | 30.72 | 29.18 | 29.25 | 29.79 | 29.50 | 30.26 | 29.29 | 29.86 |

**Table 7. Processing time (seconds) for the five fetal heartbeats in Fig 2, for the state-of-the-art chains in Table 1.**

| Tschirren et al. | Greenspan et al. | Syeda-Mahmood et al. | Magagnin et al. | Kiruthika et al. | Zolgharni et al. | Sulas et al. | MC |
|---|---|---|---|---|---|---|---|
| 1 | 1.06 | 2.32 | 0.55 | 2.84 | 0.11 | 29.84 | 29.86 |

MC. On this basis, if the goal is real-time implementation, this study represents a very preliminary assessment of the studied methods.

## Conclusion

To date, many studies of cardiac ultrasound signal processing have been reported with the aim of identifying good processing chains to achieve an optimal approximation of the Doppler velocity profile. In this work, we analyzed these studies by combining the techniques they used with a supervised classifier, i.e., a simple ANN, to identify the presence of complete and measurable fetal heartbeats from the PWD envelope. We investigated the impact of the different steps in the processing chain and compared the different techniques on the same dataset to identify a near-optimal processing chain in terms of classifier accuracy.

Although we could identify such a near-optimal processing chain and our comparison of different techniques revealed the superiority of some methods, when the aim is the recognition of complete and measurable fetal cardiac cycles within a trace, rather than the quality of the envelope itself, the role of some processing steps in the chain can have a limited effect. In fact, we found only the binarization and envelope extraction steps to have a major effect on the final result, so the greatest attention should be paid to the selection of these processing steps.

The results of this study, even in the light of its limitations, are important for researchers interested in the development of processing techniques for the automatic identification of clinically meaningful fetal heart cycles in long recordings or in scenarios where a computer aid could facilitate examination.

## Acknowledgments

The authors wish to thank the team headed by Dr. Roberto Tumbarello, Division of Pediatric Cardiology, AOB Hospital (Cagliari, Italy), for its valuable support and all the pregnant women volunteers who participated in this study for their patience and availability.

## Author Contributions

**Conceptualization:** Danilo Pani.

**Data curation:** Eleonora Sulas, Monica Urru, Giuliana Solinas.

**Funding acquisition:** Luigi Raffo, Danilo Pani.

**Investigation:** Eleonora Sulas, Emanuele Ortu.

**Methodology:** Eleonora Sulas, Emanuele Ortu, Giuliana Solinas, Danilo Pani.

**Project administration:** Danilo Pani.

**Resources:** Luigi Raffo.

**Software:** Eleonora Sulas.

**Supervision:** Roberto Tumbarello, Danilo Pani.

**Validation:** Monica Urru, Giuliana Solinas, Danilo Pani.

Writing – original draft: Eleonora Sulas, Monica Urru, Danilo Pani.

Writing – review & editing: Eleonora Sulas, Roberto Tumbarello, Luigi Raffo, Giuliana Solinas, Danilo Pani.

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
