## [Decision Letter · Decision Letter 0]

25 Jun 2020

PONE-D-20-10354

Impact of pulsed-wave-Doppler velocity-envelope extraction techniques on classification of complete fetal cardiac cycle

PLOS ONE

Dear Dr. Sulas,

Thank you for submitting your manuscript to PLOS ONE. After careful consideration, we feel that it has merit but does not fully meet PLOS ONE’s publication criteria as it currently stands. Therefore, we invite you to submit a revised version of the manuscript that addresses the points raised during the review process.

Reviewer feedback was mixed, so as a result we require you to provide substantial changes before we can accept this manuscript. Without sufficient changes, this manuscript will not be accepted.

We look forward to receiving your revised manuscript.

Kind regards,

Gordon Niall Stevenson, DPhil

Academic Editor

PLOS ONE

Additional Editor Comments:

The authors need to consider the comments provided by the reviewers and make substantial changes based on reviewer feedback for this submission to be accepted.

2. Please amend your Methods section to provide the following information:

*Please state whether data/images were anonymized prior to analysis.

*Please describe how the optimal sample size (n = 25) was determined.

*Although you provide a reference for the Matlab graphical interface used, we ask that you provide a brief description of it within the Methods section.

'Eleonora Sulas is grateful to Sardinia Regional Government for supporting her PhD scholarship (P.O.R. F.S.E., European Social Fund 2014-2020).'

'The author(s) received no specific funding for this work.'

5. Please amend either the title on the online submission form (via Edit Submission) or the title in the manuscript so that they are identical.

6. Please amend the manuscript submission data (via Edit Submission) to include author Emanuele Ortu.

7. PLOS requires an ORCID iD for the corresponding author in Editorial Manager on papers submitted after December 6th, 2016. Please ensure that you have an ORCID iD and that it is validated in Editorial Manager. To do this, go to ‘Update my Information’ (in the upper left-hand corner of the main menu), and click on the Fetch/Validate link next to the ORCID field. This will take you to the ORCID site and allow you to create a new iD or authenticate a pre-existing iD in Editorial Manager. Please see the following video for instructions on linking an ORCID iD to your Editorial Manager account: https://www.youtube.com/watch?v=_xcclfuvtxQ

8. Your ethics statement must appear in the Methods section of your manuscript. If your ethics statement is written in any section besides the Methods, please move it to the Methods section and delete it from any other section. Please also ensure that your ethics statement is included in your manuscript, as the ethics section of your online submission will not be published alongside your manuscript.

Reviewers' comments:

Reviewer's Responses to Questions

**Comments to the Author**

1. Is the manuscript technically sound, and do the data support the conclusions?

Reviewer #1: Yes

Reviewer #2: Partly

Reviewer #3: Partly

Reviewer #4: Yes

Reviewer #5: Yes

Reviewer #6: No

2. Has the statistical analysis been performed appropriately and rigorously? 

Reviewer #1: Yes

Reviewer #2: Yes

Reviewer #3: I Don't Know

Reviewer #4: Yes

Reviewer #5: Yes

Reviewer #6: No

3. Have the authors made all data underlying the findings in their manuscript fully available?

Reviewer #1: Yes

Reviewer #2: Yes

Reviewer #3: Yes

Reviewer #4: Yes

Reviewer #5: Yes

Reviewer #6: Yes

4. Is the manuscript presented in an intelligible fashion and written in standard English?

Reviewer #1: Yes

Reviewer #2: Yes

Reviewer #3: Yes

Reviewer #4: No

Reviewer #5: No

Reviewer #6: Yes

5. Review Comments to the Author

Reviewer #1: This is a well conducted study which examines the influence of various pre-processing steps on the performance of an ANN to classify fetal ultrasound beats as of sufficient quality to be processed. This is an important problem within its field, evidenced by the numerous citations.

The authors should also be commending for opening up a very large system of expertly labelled beats.

However, coming from the field of cardiology image processing, however, this choice of ANN surprises me. This appears to be an image segmentation and classification task, a task which convolutional neural networks excel at, without the requirement of such processing steps using fairly arbitrary hyperparmeters.

The use of only one ultrasound machine undoubtedly leads to questions about how well these results will generalise to other manufacturers. It's not hard to imagine the results would be completely different on e.g. a GE machine, where doppler envelopes may be on different colours, with different filters, at different resolutions.

Reviewer #2: The study investigates the impact of pulsed-wave-Doppler velocity-envelope extraction techniques on classification of complete fetal cardiac cycle.

I have the following concerns/suggestions which the authors may wish to address:

Figure 1 spans several heartbeats. Please specify to which heartbeat the corresponding waveform is shown on the right.

Some symbols are missing in equation 1

What is the input to the ANN classifier?

Please provide a schematic diagram of the neural network classifier adopted in your study and justify your choice of design. The authors have stated 'The number of neurons in the hidden layer and the number of 335 hidden layers were empirically chosen to obtain acceptable performance on the available 336 dataset without incurring an overfitting problem.' Please provide more details.

The authors state that the aim of the performance assessment was to identify the `best' processing chain. They have examined they individual steps one at a time. However, this will not guarantee the optimised combination of all processing steps will be obtained. Please reconsider your method. Have you considered all possible combinations? You have stated 'Even if the ultimate version is not the `best' chain in a strict sense, it is a reasonable approximation of it.' Can you please provide evidence for this?

The dataset corresponds to one type of machine "Philips iE33 ultrasound machine", therefore any proposed conclusion will not be vendor-neutral. Please either use data from other machines as well, or make sure you state this limitation of the study in your discussion/conclusion.

Please highlight other limitations of your study as well.

Reviewer #3: The work presented in this research article aims at finding the best signal processing chain for the extraction of velocity-envelopes of pulsed wave Doppler images acquired from healthy fetuses. Evaluation of the individual steps involved in this processing chain is an interesting and important topic to improve the assessment of fetal cardiac function. While the work presented is very relevant, this research article could strongly benefit from a more concise structure and improved representation of the results and discussion.

Please find some of the comments below:

Line 13: Explanation EAV, Early filling, active filling and V?

Line 16: Provide an image of the view chamber view ( maybe combine with Fig 1), indicating the sample volume from which the PWD image is derived.

Line 36: PWD image.

Line 67: The authors should explain why they only focus on the upper and lower envelope, which represents the peak velocity of diastolic and systolic flow. Another very common way in assessing Doppler images is to compute the mean velocity wave from which information can be derived.

Line 78: The authors should elaborate on the biggest gap algorithm. It is only later introduced.

Line 91: The Bland-Altman analysis between automatic envelope detection and manual envelope detection is probably much different than the analysis for automatic envelope detection vs. ECG signal.

Line xx: The authors may explain shortly the difference to Doppler based fetal heart rate monitoring, where instead of Doppler image a Doppler signal is directly acquired and then converted to an envelope from which the heart rate and other features can be extracted. The authors may want to refer to the articles: Voicu, Iulian, Sébastien Ménigot, Denis Kouamé, and Jean-marc Girault. 2014. “New Estimators and Guidelines for Better Use of Fetal Heart Rate Estimators with Doppler Ultrasound Devices.” Computational and Mathematical Methods in Medicine 2014. or Hamelmann P, Vullings R, Kolen AF, et al. Doppler Ultrasound Technology for Fetal Heart Rate Monitoring: A Review. IEEE Trans Ultrason Ferroelectr Freq Control. 2020;67(2):226-238. doi:10.1109/TUFFC.2019.2943626

Line 136: The description is a bit misleading and the authors should describe it more explicit. There are five phases where the authors identified the state of the art (shaded grey), but for the first phase, two methods are considered state of the art?

Line 161: In this quantitative analysis of the individual steps involved for envelope extraction, I would expect that the analysis is not based on empirical selected parameters. Instead, different sigmas should be taken. If it is not possible/feasible to do this analysis, the authors should give a better reason than just mentioning the tradeoff between noise suppression and blurring of the image. A sigma of 1 or 2 may be just as good or even better.

Line 175: check right side of equation, upside down question mark and exclamation mark in equation,

Material and Methods section: while related work is already previously discussed, in the material and methods section still different techniques are explained and introduced. For the reader it is difficult to understand which exact methods are used in this particular study. Looking at Figure 3, it suggests that there are 5 (or 6) phases identified with in total 23 different processing options. However, in the later results only 17 steps are compared.

Also, in this section more detailed information is provided on the methods already described the section related work. As consequence, a lot of things are described twice. It would be more logical to move this information into the background information section and here explain a bit more in detail how the analysis is done. A schematic block diagram of the study performed could help the reader.

A suggestion for Fig 2.: Show the result of each processing step on the PWD image. This would help the reader to better understand the differences between the individual steps.

Line 300: and likewise the highest pixel?

Fig 3: label axes. Also in the description, it misses that it shows the accuracy values. Beside that, the whole figure and subplots show the same information multiple times. The authors should consider reducing the image to subplot g and enlarging it to make the whole scale visible. Also instead of number 1-17, the respective processing steps could be written in the label. At the moment, this is very unclear. Alternatively, the whole figure could be changed to a table, also showing p-values etc.

Line 415-416: Unclear which accuracy values are referred to. Number 4,5,6,7,8,9 with respect to MC?

Line 437 - : The results are presented in a unclear and confusing way. The p-values mentioned in the text are not the one in the table. Also again, instead of labeling it with numbers and steps, use the method applied as labels.

Fig 4: Absolutely unclear what the numbers are referring to and what is shown in the image. Do i understand correctly that number 1 refers to processing chain in literature [25], number 2 to literature [8] and so on. And 8 to the MC chain and 9 to random?

463: Specify minimum expectations.

Result discussion: For the comparison of the different processing chains described in literature, I would strongly recommend using the Authors name of the method instead of the reference number. With all the numbers and steps/phases, this is extremely confusing.

Line 465: Please explain how this conclusion is derived ( 90% wider distribution), where the reader can see this, and what it means.

A critical discussion of the results are missing. E.g. what is the influence of the classifier on the performance of the processing techniques. Another classifier may lead to different "best processing chain".

What is the purpose of including the random processing chain ( and which steps are involved). The results suggest that studies 1,2,6, are even worse than a random processing chain which is hardly imaginable.

Reviewer #4: This manuscript provides a useful overview and in-depth analysis of the processing steps involved in tracing of PWD traces. The paper has discussed several different methods used in the literature as well as a step-by-step description at each stage.

General comments

Overall the content of the manuscript is sound, and the conclusions made are supported by the data. Furthermore, this analysis is useful for any researchers developing automated Doppler tracing methods as it has covered a good range of commonly used techniques and their significance to the method overall. Additionally, the method can easily be adapted for other analysis methods to determine significant steps within multi-stage automated algorithms.

The statistical analysis appears to be performed appropriately, and the authors have said the data will be made fully available in the declaration, however, the location/access to this data needs to be emphasised/clarified further in the manuscript.

The reviewer strongly suggests the authors have the manuscript reviewed by a native English speaker or alternatively avail an English editing service. There are several instances in the manuscript where the language is unclear and ambiguous.

Specific comments:

In the methods section (final paragraph), it is unclear how the datasets were classified as meaningful vs. incomplete.

Fig 4: For clarity, include a key in the legend stating which chain refers to which method in the literature.

Reviewer #5: The authors tackle the problem of PWD envelope classification. The authors propose combination of techniques that have been used in other studies in addition to the Artificial Neural Network (ANN) as a supervised classifier to obtain indications the presence of complete and measurable fetal heartbeats from the PWD envelope. They provided an experiment result with decent accuracy performance and several statistical tests. Overall, I find the paper interesting.

The paper needs a linguistic review especially abstract section and would benefit being shorter and more to the point. There are several repeating and redundant sentences.

Materials and methods section: In the “Classifier section”, an ANN has been used. It is ambiguous that what kind of artificial neural network have been used. Types of layers are included in the network and their characteristic should be specify with the details. Also, it is not clear how many samples for the train, validation and test have been used for the ANN. Perhaps, a better classifier could be used to extract the features. It would be interesting to see how it could perform for more reliable and complex Convolution Neural Network as well.

Materials and methods section (line 227): A reference or justification why for the Sobel edge detector a kernel size of 3 × 3 is used.

Fetal PWD dataset section: How many pediatric cardiologist groups were used to label the dataset. If there is only one group of cardiologists, the conclusion in this manuscript could not be supported strongly.

The figure 2 Envelope Extraction Workflow is not clear, it is highly recommended to integrate with picture/figure to demonstrate the procedure for each step. It is very difficult to follow the workflow.

Computation time is important specially in the classification task. In total how long does the process takes compared to other works.

How this proposed method would perform on private/public dataset or dataset from different site or machine?

Additional remarks:

• Page 2 (line 48, 49) The section relevant to the classes and images in introduction section could be placed in the Fetal PWD dataset section. Also, in the introduction the paragraph “custom dataset of fetal PWD 47 recordings acquired from healthy fetuses at the Division of Pediatric Cardiology, San 48 Michele Hospital, Cagliari, Italy” is redundant and already explained in the “Fetal PWD dataset” section.

• Page 5 (line 130), In the “material and method” section there is no information about the dataset, perhaps the title of this section could be “method”.

• For fig 1 provide the extend form for E, A and V (what they are stand for?)

• Please correct the format in equation (1)

• Please add more parameters' definitions in equation (2), what are grandma and grandmin

• please correct the format in equation (9), × 100 is missing.

• In Performance assessment section, the paragraph “To assess the symmetry of the distributions, we adopted 350 box-and-whiskers plots” please refer to the figure number.

• Refences in the text are not in order. For example, your first reference is [28] in introduction section.

I look forward to see how the algorithm will perform when test on samples with the different heart conditions.

Reviewer #6: In this work, the authors compare between different pulsed wave doppler envelope extraction pipelines by analyzing the different steps involved in the pipeline. They use a downstream classification task accuracy to compare between the signal processing steps involved in the pipeline and recommend important steps in the envelope extraction pipeline.

1. This manuscript introduces very little to no novel methodology and the work is too incremental.

2. The downstream validation was done on a single classification task using a trained neural network which is insufficient for a lot of claims made in the manuscript. Having supporting validation on multiple downstream detection/classification tasks would add more value to the contribution.

6. PLOS authors have the option to publish the peer review history of their article (what does this mean?). If published, this will include your full peer review and any attached files.

Reviewer #1: No

Reviewer #2: No

Reviewer #3: No

Reviewer #4: No

Reviewer #5: No

Reviewer #6: No

---

## [Author Response · Author response to Decision Letter 0]

3 Jan 2021

Reviewer #1: This is a well conducted study which examines the influence of various pre-processing steps on the performance of an ANN to classify fetal ultrasound beats as of sufficient quality to be processed. This is an important problem within its field, evidenced by the numerous citations.

The authors should also be commending for opening up a very large system of expertly labelled beats.

>> We thank the Reviewer for appreciating the methodology of our study and the efforts in annotating the data that will be made publicly available.

However, coming from the field of cardiology image processing, however, this choice of ANN surprises me. This appears to be an image segmentation and classification task, a task which convolutional neural networks excel at, without the requirement of such processing steps using fairly arbitrary hyperparmeters.

>> We thank the Reviewer for this comment and to give us the opportunity to explain this preference. The following points should be carefully considered:

1. the clinical analysis is routinely carried out by the cardiologists by visually extracting the PWD velocity envelopes from the images, which can be used for other clinical tasks such as measurement of intervals,

2. in previous work [1] we demonstrated how a supervised classifier trained with the samples representing the upper and lower envelopes of the PWD, with additional features extracted from the image, achieved significantly better results (p < 0.0001) than other algorithms based on other reasoned features,

3. focusing on the importance of each of the preprocessing steps involved in the extraction of the PWD envelopes could then represent an interesting point, considering that the literature in the field is continuously proposing new methods for the envelope extraction,

4. since the envelope has a memory footprint dramatically smaller compared to the whole image, using it to produce features rather than the image has obvious advantages for the potential implementation of the algorithm in the device,

5. the choice of the ANN has been also supported by our previous results [2] and [3], where the performance for this specific problem were excellent,

6. since the computational complexity of a conventional ANN is dramatically lower than that of a CNN, the former appears to be a better option than the latter for implementation in the US device,

7. the dataset size for a CNN should be preferably larger than the available one, considering the number of parameters to set.

We think this motivates our choice and the rationale behind our work. By elaborating on the Reviewer’s comment, we better discussed this aspect in the revised version of the manuscript.

The use of only one ultrasound machine undoubtedly leads to questions about how well these results will generalise to other manufacturers. It’s not hard to imagine the results would be completely different on e.g. a GE machine, where doppler envelopes may be on different colours, with different filters, at different resolutions.

>> We agree with the Reviewer that the adoption of a single device is a limitation but it is actually a minor one. In fact, for sure different ultrasound machines can present different axes, grid colors, physio traces, etc… however, these aspects concern the extraction of the single wide-image from the video, and then a step before the studied processing steps and not considered in this work (it is common for all the methods and any change on this could only lead to a common bias). For sure, video resolutions can be also different from device to device so that the number of pixels that compose a second in the video image (and then the sampling frequency of the extracted envelopes) could be different. However, the differences between different devices is limited and again this simply represents a uniform bias for the tested methods. Moreover, the size of the selected window for the image crop can be changed to meet the specification of the device in use. About the filters, there are no filters applied on the images or in any case they are set by the clinician for the best (human) view. Moreover, in our study, we also demonstrate that the first preprocessing steps, which exploited image filtering to enhance the contrast, do not play an important role in the classification output. Therefore, it is reasonable to believe that different filters from different manufacturers will not have a big impact on the envelope extraction. About the envelope colors, we cannot see the point since the envelope is extracted by the studied methods and the colors present in Fig. 2 were only used to highlight the extracted envelope but they are not originally in the image obtained by the US device. Finally, for sure the definition and contrast of the images can change from manufacturer to manufacturer but even the mean pixel values of the four important areas in the selected window (used as features for the ANN) cannot assume a manufacturer-specific behavior. In fact, if the component of the fetal heartbeats is high, the mean pixel brightness will also be higher. This is a common characteristics of the PWD traces, regardless of the manufacturer: if this would not be true, also the diagnostic process would be severely affected by the differences, which is not true.

Overall, this limitation (which is common to all the studies in the literature cited in our work) is reasonably of minor impact on the results, also because it would only produce a bias common to all the tested tecniques. Moreover, for a complete analysis as suggested by the Reviewer, the effect of the machines should be evaluated on the same subjects, and this would introduce several issues including lack of instrumentation, availability of volunteers, clinicians’ time, and violation of the experimental protocol approved by the ethics committee of the Hospital which does not foresee additional examination on the pregnant women.

According to the Reviewer’s comment, we better discussed these aspects it in revised version of the manuscript, clearly reporting the study limitation.

[1] E. Sulas, M. Urru, R. Tumbarello, L. Raffo, and D. Pani, “Automatic Detection of Complete and Measurable Cardiac Cycles in Antenatal Pulsed-Wave Doppler Signals,” Comput. Methods Programs Biomed., p. 105336, 2020.

[2] E. Sulas, E. Ortu, L. Raffo, M. Urru, R. Tumbarello, and D. Pani, “Automatic Recognition of Complete Atrioventricular Activity in Fetal Pulsed-Wave Doppler Signals,” in 2018 40th Annual International Conference of the IEEE Engineering in Medicine and Biology Society (EMBC), 2018, pp. 917–920.

[3] D. Pani, E. Sulas, E. Ortu, M. Urru, A. Cadoni, R. Tumbarello, L. Raffo, “Fetal Pulsed-Wave Doppler Atrioventricular Activity Detection by Envelope Extraction and Processing,” in Computing in cardiology, 2018.

 

Reviewer #2: The study investigates the impact of pulsed-wave-Doppler velocity-envelope extraction techniques on classification of complete fetal cardiac cycle. I have the following concerns/suggestions which the authors may wish to address:

Figure 1 spans several heartbeats. Please specify to which heartbeat the corresponding waveform is shown on the right.

>> We are sorry for this missing information. Figure 1 was composed of three images: a first long images with several fetal beats and the traced envelopes, a single fetal PWD beat belonging to another trace and a sketch of a single beat, which was drawn to explain, to a non-expert reader, the naming of the single waves composing it as used throughout the manuscript. Considering the Reviewer’s request and the clearness of that image, we modified the Figure 1 as follow: the PWD of the first five fetal heartbeats of the original PWD image (named PWD image 1 in our dataset that will be publicly shared in a Mendeley data repository) and then the sketch of one single beat. We think this is better for the reader.

Some symbols are missing in equation 1.

>> We thank the Review for pinpointing this issue. We correct the mistake.

What is the input to the ANN classifier?

>> As described in Performance Assessment section, in this work, the ANN is feed by an array of 264 input features are chosen, 128 samples of the upper envelope 𝐺𝑢 and 128 of the lower envelope 𝐺𝑙 (normalized respectively between 0 and 1 and between -1 and 0), 4 area features (area under the envelope) and 4 pixel features (mean values of the pixels under the envelopes). The four area and pixel features were extracted from the 4 quadrants of a windowed image, whose length was chosen equal to 128 samples by taking into account that (at 248 Hz of sampling frequency for the two envelopes) a healthy fetus at that gestation age (between the 21st to the 27th) could have a heart rate ranging from 110 to 160 bpm, so that a length of 128 samples typically is able to represent a whole beat.

According to the Reviewer’s comment, we better explained this in the manuscript, adding also an image regarding the features and the used classifier.

Please provide a schematic diagram of the neural network classifier adopted in your study and justify your choice of design. The authors have stated 'The number of neurons in the hidden layer and the number of hidden layers were empirically chosen to obtain acceptable performance on the available dataset without incurring an overfitting problem.' Please provide more details.

>> According to the Reviewer’s request, we added a schematic diagram of the ANN adopted in this work, along with an extended description of the reason why we adopted the ANN and how. By using the scaled conjugate gradient back-propagation algorithm, we trained an ANN characterised by 264 input nodes, depending on the number of extracted features, 10 hidden nodes and two output nodes, which is equal to the number of classes that we wanted to identify.

We sincerely thank the Reviewer for highlighting this issue. About the chosen model, in previous works [1]–[3], it allowed achieving the best performance with a reduced risk of overfitting, so that paramterization was chosed also in this case.

The authors state that the aim of the performance assessment was to identify the `best' processing chain. They have examined they individual steps one at a time. However, this will not guarantee the optimised combination of all processing steps will be obtained. Please reconsider your method. Have you considered all possible combinations? You have stated 'Even if the ultimate version is not the `best' chain in a strict sense, it is a reasonable approximation of it.' Can you please provide evidence for this?

>> We thank the Review for this comment. The chosen approach is commonly used in statical analysis when considering all the possible combinations of the parameters is unfeasible. For sure, naming it the ‘best’ chain can also be seen as inappropriate, so we reworded in “near-optimal”, also thanks to the consultation with the professional editors who managed the proofreading of the manuscript.

The dataset corresponds to one type of machine "Philips iE33 ultrasound machine", therefore any proposed conclusion will not be vendor-neutral. Please either use data from other machines as well, or make sure you state this limitation of the study in your discussion/conclusion.

>> We agree with the Reviewer that the adoption of a single device is a limitation but it is actually a minor one. In fact, for sure different ultrasound machines can present different axes, grid colors, physio traces, etc… however, these aspects concern the extraction of the single wide-image from the video, and then a step before the studied processing steps and not considered in this work (it is common for all the methods and any change on this could only lead to a common bias). For sure, video resolutions can be also different from device to device so that the number of pixels that compose a second in the video image (and then the sampling frequency of the extracted envelopes) could be different. However, the differences between devices is limited and again this simply represents a uniform bias for the tested methods. Moreover, the size of the selected window for the image crop can be changed to meet the specification of the device in use. About the filters, there are no filters applied on the images or in any case they are set by the clinician for the best (human) view. Moreover, in our study, we also demonstrate that the first preprocessing steps, that exploited image filtering to enhance the contrast, do not play an important role in the classification output. Therefore, it is reasonable to believe that different filters from different manufacturers will not have a big impact on the envelope extraction. Finally, for sure the definition and contrast of the images can change from manufacturer to manufacturer but even the mean pixel values of the four important areas in the selected window (used as features for the ANN) cannot assume a manufacturer-specific behavior. In fact, if the component of the fetal heartbeats is high, the mean pixel brightness will also be higher. This is a common characteristics of the PWD traces, regardless of the manufacturer: if this would not be true, also the diagnostic process would be severely affected by the differences, which is not true.

Overall, this limitation (which is common to all the studies in the literature cited in our work) is reasonably of minor impact on the results, also because it would only produce a bias common to all the tested tecniques. Moreover, the effect of the machines should be evaluated on the same subjects, and this would introduce several issues including lack of instrumentation, availability of volunteers, clinicians’ time, and violation of the experimental protocol approved by the ethical committee of the Hospital which does not foresee additional examination on the pregnant women.

According to the Reviewer’s comment, we better discussed these aspects it in revised version of the manuscript, clearly reporting the study limitation.

Please highlight other limitations of your study as well.

>> We thank the Review for this comment. As we already mentioned before, we added a Discussion section where we better examine also the limitation of this work. 

Reviewer #3: The work presented in this research article aims at finding the best signal processing chain for the extraction of velocity-envelopes of pulsed wave Doppler images acquired from healthy fetuses. Evaluation of the individual steps involved in this processing chain is an interesting and important topic to improve the assessment of fetal cardiac function. While the work presented is very relevant, this research article could strongly benefit from a more concise structure and improved representation of the results and discussion.

>> We thank the Reviewer for the useful suggestions we exploited to improve the quality of the manuscript.

Please find some of the comments below:

Line 13: Explanation EAV, Early filling, active filling and V?

>> We thank the Reviewer for pinpointing this lacking information. The image includes the upper (blue line) and lower (red line) envelopes. On the right, a graphical representation of the fetal beat waveform morphology acquired with the PWD technology using an apical 5-chambers window is shown. The typical patter consists of two main waves and associated three peaks: E, that refers to the early, passive diastolic filling (E stands for “early”), A, associated with the late, diastolic active ventricular filling by the atrial contraction (A stands for “atrial”), and V, that represents the ventricular flow peak (V stands for “ventricular”. We clarified this aspect in the first section of the manuscript.

Line 16: Provide an image of the view chamber view (maybe combine with Fig 1), indicating the sample volume from which the PWD image is derived.

>> Added.

Line 36: PWD image.

>> Fixed.

Line 67: The authors should explain why they only focus on the upper and lower envelope, which represents the peak velocity of diastolic and systolic flow. Another very common way in assessing Doppler images is to compute the mean velocity wave from which information can be derived.

>> We thank the Reviewer for this important comment. In our previous works, we found that the performance of an ANN who exploits as inputs the lower and the upper envelope of the PWD information are adequate to discriminate between complete and clinically meaningful fetal heartbeats and uncomplete/noisy/malformed ones. The focus on the velocity profile descends from the clinical experience of the authors, who exploit them for the fetal heart assessment and for the purposes of this work. Conversely, the mean velocity is not exploited by them to identify the complete and measurable beats even though in the future we will try embedding this information in the classification process.

Based on the focus on the upper and lower envelopes, that already provided excellent results also in previous works, in this work we investigated the envelope extraction process by looking at the impact of the various processing techniques already presented in the literature in terms of our classification output. A rigorous assessment in this sense is absolutely missing in the scientific literature. In the revised version of the manuscript, we further motivated this choice in the Introduction by saying that “Compared to the use of synthetic indexes such as the mean velocity or other parameters that can be extracted from the Doppler image, by focusing only on the PWD envelope it is possible to reduce the impact of computational errors occurring in their extraction. Moreover, this choice closely mimics the usual clinical experience in the identification of well-formed PWD traces with a clinical value for the subsequent analyses.”

In the revisied version of the manuscript, we briefly added in the Introduction some of the main indexes used by clinicians for the assessment of the PWD image. All of them can be obtained from the velocity envelope, even though this further step is out of the scope of our manuscript.

Line 78: The authors should elaborate on the biggest gap algorithm. It is only later introduced.

>> According to the Reviewer’s comment, we added that this was a custom algorithm that will be discussed in the next section along with all the other processing steps. In fact, presenting in section 2 this method would require to re-think completely the structure of sections 2 and 3.

Line 91: The Bland-Altman analysis between automatic envelope detection and manual envelope detection is probably much different than the analysis for automatic envelope detection vs. ECG signal.

>> We thank the Reviewer for this comment. The Section aimed to give a brief introduction to the works composing the state-of-the-art. In that line, we were referring to the work [4], where they extract the envelopes and then fit a parametric model to the fetal heartbeat. They were able to do it because of the presence of the annotation of the fetal heartbeats obtained from the ECG data. The results were shown in terms of errors between the clinical parameters computed on the automated extracted maximal velocity envelope and the manually-extracted ones. Beat-by-beat comparison, as well as average beat comparison between the automated measurements and the technicians, were performed in [4], and the relative errors were measured by Bland–Altman analysis.

We better explained this aspect in the revised version of the manuscript.

Line xx: The authors may explain shortly the difference to Doppler based fetal heart rate monitoring, where instead of Doppler image a Doppler signal is directly acquired and then converted to an envelope from which the heart rate and other features can be extracted. The authors may want to refer to the articles: Voicu, Iulian, Sébastien Ménigot, Denis Kouamé, and Jean-marc Girault. 2014. “New Estimators and Guidelines for Better Use of Fetal Heart Rate Estimators with Doppler Ultrasound Devices.” Computational and Mathematical Methods in Medicine 2014. or Hamelmann P, Vullings R, Kolen AF, et al. Doppler Ultrasound Technology for Fetal Heart Rate Monitoring: A Review. IEEE Trans Ultrason Ferroelectr Freq Control. 2020;67(2):226-238. doi:10.1109/TUFFC.2019.2943626

>> We thank the Reviewer for this suggestion. However, that modality is not included in the clinical tools used in the hospital for fetal heart assessment. Moreover, several reviewers complained about the manuscript length (which unfortunately increased more because of the asked revisions) and this interesting point is marginal for the understanding of the manuscript content, so we apologize but we think it could not find a place in the revised version.

Line 136: The description is a bit misleading and the authors should describe it more explicit. There are five phases where the authors identified the state of the art (shaded grey), but for the first phase, two methods are considered state of the art?

>> We thank the Reviewer for this comment. Yes, the main phases are five, but the first phase can also be composed by two concatenate processing steps instead of only one. In order to help the reader in a smooth understanding of the text and figures, we modified the (currently) figure 3 to group under the same brackets the first two blocks.

Line 161: In this quantitative analysis of the individual steps involved for envelope extraction, I would expect that the analysis is not based on empirical selected parameters. Instead, different sigmas should be taken. If it is not possible/feasible to do this analysis, the authors should give a better reason than just mentioning the tradeoff between noise suppression and blurring of the image. A sigma of 1 or 2 may be just as good or even better.

>> We thank the Reviewer for this comment. The aim of this work is to compare the different possible workchain that are possible for the extraction of the envelope from the PWD images. In this work, we specifically compared the previous studies and implemented algorithm, that are schematized in the Table 1 in the manuscript. The Gaussian filter with a sigma = 1.5 was found to be the best trade-off in the three previous works that exploited that filter. It is out of the scope to finetune the parameters already studied in the previous studies. However, our aim is not to give only one near-optimal processing workchain for envelope tracing in terms of the fetal beats classification, but it is also to guide the researchers that are facing a similar problem, to focus on the processing steps that have the highest impact on the output. In this case, if the Gaussian filter have resulted to be very important, I would also aspect that the researchers interested in the field will try to find the best trade off in terms of sigma and obtained results, instead of paying more attention to a less important step. We better clarify this aspect in our work, and explain it in the discussion section.

Line 175: check right side of equation, upside down question mark and exclamation mark in equation,

>> We thank the Reviewer. We corrected this mistake.

Material and Methods section: while related work is already previously discussed, in the material and methods section still different techniques are explained and introduced. For the reader it is difficult to understand which exact methods are used in this particular study. Looking at Figure 3, it suggests that there are 5 (or 6) phases identified with in total 23 different processing options. However, in the later results only 17 steps are compared.

>> We thank the Reviewer. We first discuss the state-of-the-art studies in their respective context and aim. Then, in the Material and methods section, we briefly describe the different steps that can be used in each processing phase, so phase-wise and not method-wise. This is because these steps are used in the work even outside the context of the original envelope tracing methods. We tried to better clarify the approach in the revised version of the manuscript.

About the second part of the comment, we thank again the Reviewer for the precious indication. We have 23 possible processing steps involved in the envelope tracing. However, for the first part of our assessment, the main work chain is modified by substituting only one processing step at a time with the remaining available for that phase. Since the MC is composed of 6 steps, the different chains composed of 5 steps belonging to the MC and one step from the pool of the available ones will be 23-6=17. We tried to better explain this aspect in the revised version of the manuscript, also by adding a figure.

Also, in this section more detailed information is provided on the methods already described the section related work. As consequence, a lot of things are described twice. It would be more logical to move this information into the background information section and here explain a bit more in detail how the analysis is done. A schematic block diagram of the study performed could help the reader.

>> We thank the Reviewer for this suggestion. In the manuscript, we worked on improving the discussion of the related works and the material and methods contents. We hope that the manuscript is now clearer. The main idea, however, is to present in the related works the techniques used and developed for envelope tracing, in the contexts envisioned by the authors, and in the Materials and Methords section a detailed technical description of the processing steps studied in this work. In the latter, we provided a technical description of the processing steps of the algorithms identified in the previous section, which were used in our systematic analysis presented here, even when they are outside the context of the original algorithms. We better explained these aspects in the manuscript.

Moreover, according to the Reviewer’s comment, we added more figures and tables to ease the reading.

A suggestion for Fig 2.: Show the result of each processing step on the PWD image. This would help the reader to better understand the differences between the individual steps.

>> We thank the Reviewer. Hereafter we report six images that represents the results of each processing step. Each image consists of 18 images, where, drawn in the image, the red line is the upper envelope, blue line is the lower envelop. There are 3 images per row for each of the 6 images. The first row shows (from left to right): (1) MC with no filter application for the first preprocessing step; (2) MC using as an intensity adjustment the subtraction of the greyscale and modified images; (3) MC using no operation for the second preprocessing step. The second row shows (from left to right): (4) MC using Canny algorithm; (5) MC using NLLAP edge detector (NL); (6) MC using a combination of the NLLAP and Sobel edge detectors. The third row shows (from left to right): (7) MC using an adaptive threshold; (8) MC using the steepest gradient histogram threshold; (9) MC using the Otsu threshold. The forth row shows (from left to right): (10) MC using the region-growing algorithm; (11) MC removing spurious areas using a morphological operation (MO) with maximum 500-connected pixels; (12) MC removing spurious areas using a morphological operation (MO) with maximum 50-connected pixels. The fifth row shows (from left to right): (13) No post-binarisation processing; (14) MC using the biggest-gap algorithm; (15) MC using median filter of 5-points length. The last row shows (from left to right): (16) MC using median filter of 15-points length (17) MC using a low-pass first-order Butterworth filter; (18) MC.

The six images are related to one of the main 6 exploited image/envelope processings. The first image is related to the image preprocessing 1, the second image to the image preprocessing 2, third image shows the results of the image binarisation, the forth image the results of the image post-processing, the fith image in the supplementary material illustrates the 18 images results from the envelope extraction, and the sex image, that we placed also in the manuscript, show the final results, so also after the envelope post-processing.

However, in the manuscript, only the the image representing the results of the tracing with the different chains was added, for a problem of space.

Figure 1

Figure 2

 

Figure 3

 

Figure 4

 

Figure 5

 

Figure 6

Line 300: and likewise the highest pixel?

>> It means considering the highest or the lowest pixel depends on which envelope is going to be extracted if the lower or the upper respectively. We improved the quality of the language for a better understanding.

Fig 3: label axes. Also in the description, it misses that it shows the accuracy values. Beside that, the whole figure and subplots show the same information multiple times. The authors should consider reducing the image to subplot g and enlarging it to make the whole scale visible. Also instead of number 1-17, the respective processing steps could be written in the label. At the moment, this is very unclear. Alternatively, the whole figure could be changed to a table, also showing p-values etc.

>> We thank the Reviewer for this comment. We improve the figure caption, the figure itself and added a table. The results for the statistical analysis are shown in table 3. The second row of the table shows the Kruskal–Wallis test result whereas the third row reports with a * any statistically significant difference in the pairwise Wilcoxon signed rank tests between the MC and the other processing chains differing only for the exploitation of another options in that step.

Line 415-416: Unclear which accuracy values are referred to. Number 4,5,6,7,8,9 with respect to MC?

>> We thank the Reviewer. We improved the way we referred to each result shown in the figures to make the manuscript clearer.

Line 437 - : The results are presented in a unclear and confusing way. The p-values mentioned in the text are not the one in the table. Also again, instead of labeling it with numbers and steps, use the method applied as labels.

>> We thank the Reviewer. We carefully reviewed the Result section to improve it.

Fig 4: Absolutely unclear what the numbers are referring to and what is shown in the image. Do i understand correctly that number 1 refers to processing chain in literature [25], number 2 to literature [8] and so on. And 8 to the MC chain and 9 to random?

>> We thank the Review for this comment. Exactly, number 1 refers to processing chain in literature [25], number 2 to literature [8], … 8 to the MC chain and 9 to random. In the revised version, we improved the way we refer to the different compared chains, explicitly referring to the exploited state-of-the-art studies. We hope that the manuscript is now clearer.

463: Specify minimum expectations.

Result discussion: For the comparison of the different processing chains described in literature, I would strongly recommend using the Authors name of the method instead of the reference number. With all the numbers and steps/phases, this is extremely confusing.

>> We thank the Reviewer for this suggestion that we followed in the manuscript. The minimum value for the accuracy considered as acceptable in this work was 90%, as now reported in the materials and methods section.

Line 465: Please explain how this conclusion is derived (90% wider distribution), where the reader can see this, and what it means.

>> In the old Line 465, we mentioned that ‘Even though the techniques presented in [23] and [11] achieved higher median accuracies than the worst chains, above 90%, their distributions are wider.’ We are referring to the previous Figure 4. In fact, we see that 5 accuracy distributions reached median values above 90%. Between these 5 distributions, we can assert that 2 of them, related to the previously cited studies, show a wider distribution, leading to higher variability of those results.

We thank the Reviewer for this comment and we made this statement clearer in the manuscript.

A critical discussion of the results are missing. E.g. what is the influence of the classifier on the performance of the processing techniques. Another classifier may lead to different "best processing chain".

What is the purpose of including the random processing chain ( and which steps are involved). The results suggest that studies 1,2,6, are even worse than a random processing chain which is hardly imaginable.

>> We thank the Reviewer. We add a new Discussion Section in the manuscript to better examine and explain the achieved result. We hope that now the manuscript is clearer.

[1] E. Sulas, M. Urru, R. Tumbarello, L. Raffo, and D. Pani, “Automatic Detection of Complete and Measurable Cardiac Cycles in Antenatal Pulsed-Wave Doppler Signals,” Comput. Methods Programs Biomed., p. 105336, 2020.

[2] E. Sulas, E. Ortu, L. Raffo, M. Urru, R. Tumbarello, and D. Pani, “Automatic Recognition of Complete Atrioventricular Activity in Fetal Pulsed-Wave Doppler Signals,” in 2018 40th Annual International Conference of the IEEE Engineering in Medicine and Biology Society (EMBC), 2018, pp. 917–920.

[3] D. P. Eleonora Sulas, Emanuele Ortu, Monica Urru, Alessandra Cadoni, Roberto Tumbarello, Luigi Raffo, “Fetal Pulsed-Wave Doppler Atrioventricular Activity Detection by Envelope Extraction and Processing,” in Computing in cardiology, 2018.

[4] H. Greenspan, O. Shechner, M. Scheinowitz, and M. S. Feinberg, “Doppler echocardiography flow-velocity image analysis for patients with atrial fibrillation,” Ultrasound Med. Biol., vol. 31, no. 8, pp. 1031–1040, Aug. 2005.

[5] E. Sulas, E. Ortu, L. Raffo, M. Urru, R. Tumbarello, and D. Pani, “Automatic Recognition of Complete Atrioventricular Activity in Fetal Pulsed-Wave Doppler Signals.,” Conf. Proc. ... Annu. Int. Conf. IEEE Eng. Med. Biol. Soc. IEEE Eng. Med. Biol. Soc. Annu. Conf., vol. 2018, pp. 917–920, Jul. 2018.

[6] E. Sulas et al., “Fetal Pulsed-Wave Doppler Atrioventricular Activity Detection by Envelope Extraction and Processing,” in Computing in Cardiology - vol. 45, 2018.

 

Reviewer #4: This manuscript provides a useful overview and in-depth analysis of the processing steps involved in tracing of PWD traces. The paper has discussed several different methods used in the literature as well as a step-by-step description at each stage.

General comments

Overall the content of the manuscript is sound, and the conclusions made are supported by the data. Furthermore, this analysis is useful for any researchers developing automated Doppler tracing methods as it has covered a good range of commonly used techniques and their significance to the method overall. Additionally, the method can easily be adapted for other analysis methods to determine significant steps within multi-stage automated algorithms.

The statistical analysis appears to be performed appropriately, and the authors have said the data will be made fully available in the declaration, however, the location/access to this data needs to be emphasised/clarified further in the manuscript.

>> We thank the Reviewer for his/her time to review our manuscript and we are glad she/he considered our study interesting.

The reviewer strongly suggests the authors have the manuscript reviewed by a native English speaker or alternatively avail an English editing service. There are several instances in the manuscript where the language is unclear and ambiguous.

>> We thank the Reviewer for this suggestion. The revised article underwent proofreading by a professional editing service (ENAGO).

Specific comments:

In the methods section (final paragraph), it is unclear how the datasets were classified as meaningful vs. incomplete.

We thank the Reviewer for this comment. We improved the description and we also added an image that shows the example of two fetal heartbeats, one representing “complete and measurable”, and one that shows an incomplete fetal cycle. The “complete and measurable” fetal heart beats are represented by the window where the atrial and the ventricular activity appear clearly by visual inspection. Moreover, the atrial activity should consist of distinguishable A and E waves.

Fig 4: For clarity, include a key in the legend stating which chain refers to which method in the literature.

We thank the Reviewer for this suggestion. We improved the figure caption to make it clearer.

 

Reviewer #5: The authors tackle the problem of PWD envelope classification. The authors propose combination of techniques that have been used in other studies in addition to the Artificial Neural Network (ANN) as a supervised classifier to obtain indications the presence of complete and measurable fetal heartbeats from the PWD envelope. They provided an experiment result with decent accuracy performance and several statistical tests. Overall, I find the paper interesting.

>> We thank the Reviewer for the time spent in reviewing our manuscript and we are glad she/he considered our study interesting.

The paper needs a linguistic review especially abstract section and would benefit being shorter and more to the point. There are several repeating and redundant sentences.

>>We thank the Reviewer. We improved the presentation and sent the revised manuscript to ENAGO for professional proofreading.

Materials and methods section: In the “Classifier section”, an ANN has been used. It is ambiguous that what kind of artificial neural network have been used. Types of layers are included in the network and their characteristic should be specify with the details. Also, it is not clear how many samples for the train, validation and test have been used for the ANN. Perhaps, a better classifier could be used to extract the features. It would be interesting to see how it could perform for more reliable and complex Convolution Neural Network as well.

>> We thank the Reviewer for this comment and to give us the opportunity to explain this preference. The adoption of the following points should be carefully considered:

1. the clinical analysis is routinely carried out by the cardiologists by visually extracting the PWD velocity envelopes from the images, which can be used for other clinical tasks such as measurement of intervals,

2. in previous work [1] we demonstrated how a supervised classifier trained with the samples representing the upper and lower envelopes of the PWD, with additional features extracted from the image, achieved significantly better results (p < 0.0001) than other algorithms based on other reasoned features,

3. focusing on the importance of each of the preprocessing steps involved in the extraction of the PWD envelopes could then represent an interesting point, considering that the literature in the field is continuously proposing new methods for the envelope extraction,

4. since the envelope has a memory footprint dramatically smaller compared to the whole image, using it to produce features rather than the image has obvious advantages for the potential implementation of the algorithm in the device,

5. the choice of the ANN has been also supported by our previous results [2] and [3], where the performance for this specific problem were excellent,

6. since the computational complexity of a conventional ANN is dramatically lower than that of a CNN, the former appears to be a better option than the latter for implementation in the US device,

7. the dataset size for a CNN should be preferably larger than the available one, considering the number of parameters to set.

We think this motivates our choice and the rationale behind our work. By elaborating on the Reviewer’s comment, we better discussed this aspect in the revised version of the manuscript.

Materials and methods section (line 227): A reference or justification why for the Sobel edge detector a kernel size of 3 × 3 is used.

>>We thank the Reviewer for highlighting this issue. There is a misunderstanding due to our unclear description in the section.

In the Materials and Methods section we aimed to give a brief theoretical explanation headed towards the Methods that we compared in our study. Therefore, we shortly described only the techniques used in the state-of-the-art studied that we described in the section ‘Related works’.

Sobel edge detector was never used in our study, but the study [4] exploited three different edge detectors: (1) Sobel, (2) NLLAP, and (3) Sobel+NLLAP technical. Their results showed that the combination of the Sobel-edge image and NLLAP-edge image was the most performant.

The sentence ‘The Sobel edge detector is a gradient-based method that uses first-order derivatives approximated using 3 x 3 kernels.’ refers to the fact that the Sobel edge detector is a gradient based method and it works with first order derivatives separately for the x and y axes. In the cited work, the kernel size for the Sobel edge detector was not mentioned, so the original size (3x3) was adopted in our study. We better clarified this point in the revised version of the manuscript.

Fetal PWD dataset section: How many pediatric cardiologist groups were used to label the dataset. If there is only one group of cardiologists, the conclusion in this manuscript could not be supported strongly.

>> Considering the presented results, because of the labelling performed by a single group of cardiologists, in principle, the results could be biased. However, no measurements were performed by the cardiologists but only visual inspections of the heart cycles: according to this, based on the current clinical knowledge and the healthy status of the fetuses, the bias introduced is negligible from a clinical perspective.

For the sake of clariness, we clarified this aspect in the manuscript in the new Discussion section.

The figure 2 Envelope Extraction Workflow is not clear, it is highly recommended to integrate with picture/figure to demonstrate the procedure for each step. It is very difficult to follow the workflow.

>> We thank the Reviewer. We improved the description of the procedure we exploited adding a new image that shows some of the created chain used in this work.

Computation time is important specially in the classification task. In total how long does the process takes compared to other works.

>> The focus of this work is to evaluate the importance of each single step that leads to the extraction of the envelope from the PWD image for the recognition of the fetal beats. This is done, for the moment, offline. In fact, we would like to study the real performing value of the processing in this problem. Therefore, the computational time effort was not consider in our study and in the evaluation of processing chains.

To compare the computational costs, the algorithm should also be optimized, in the same way, and executed on a reference architecture (MATLAB is not the appropriate environment to perform these analyses). Anyway, for the sake of clarity, as per the Reviewer question, hereafter a table that shows the time (in second) taken from the same PC to process the short dopper image that is placed in the manuscript and contains 5 fetal heart PWD cycles. The Table shows how the computational effort is mostly due to the binarization step (changed between 4 5 6 7 8 9 and MC processing chains). However those values cannot be one of the parameter to evaluate our results, but we added this information in the mauscript. The second table illustrates the comparison of the time regarding the related works and the main chain (MC). Both tables represent the results in seconds.

1 2 3 4 5 6 7 8 9 10 11 12 13 14 15 16 17 MC

26.32 30.43 30.63 0.18 0.44 0.51 0.14 0.12 0.10 32.34 30.72 29.18 29.25 29.79 29.50 30.26 29.29 29.86

Tschirren et al. Greenspan et al. Syeda-Mahmood et al. Magagnin et al. Kiruthika et al. Zolgharni et al. Sulas et al. MC

1 1.06 2.32 0.55 2.84 0.11 29.84 29.86

In the future, if the goal would be to be implemented in real time and with certain requirement constraints, this study can be a guide the evaluate the different implementation possibilities also considering to optimize all the algorithm in the same way.

We discuss this aspect in the Discussion section in the manuscript.

How this proposed method would perform on private/public dataset or dataset from different site or machine?

>> We agree with the Reviewer that the adoption of a single device is a limitation but it is actually a minor one. In fact, for sure different ultrasound machines can present different axes, grid colors, physio traces, etc… however, these aspects concern the extraction of the single wide-image from the video, and then a step before the studied processing steps and not considered in this work (it is common for all the methods and any change on this could only lead to a common bias). For sure, video resolutions can be also different from device to device so that the number of pixels that compose a second in the video image (and then the sampling frequency of the extracted envelopes) could be different. However, the differences between different devices is limited and again this simply represents a uniform bias for the tested methods. Moreover, the size of the selected window for the image crop can be changed to meet the specification of the device in use. About the filters, there are no filters applied on the images or in any case they are set by the clinician for the best (human) view. Moreover, in our study, we also demonstrate that the first preprocessing steps, that exploited image filtering to enhance the contrast, do not play an important role in the classification output. Therefore, it is reasonable to believe that different filters from different manufacturers will not have a big impact on the envelope extraction. About the envelope colors, we cannot see the point since the envelope is extracted by the studied methods and the colors present in the different figure, and for istance in Fig. 2, were only used to highlight the extracted envelope but they are not originally in the image obtained by the US device. Finally, for sure the definition and contrast of the images can change from manufacturer to manufacturer but even the mean pixel values of the four important areas in the selected window (used as features for the ANN) cannot assume a manufacturer-specific behavior. In fact, if the component of the fetal heartbeats is high, the mean pixel brightness will also be higher. This is a common characteristics of the PWD traces, regardless of the manufacturer: if this would not be true, also the diagnostic process would be severely affected by the differences, which is not true.

Overall, this limitation (which is common to all the studies in the literature cited in our work) is reasonably of minor impact on the results, also because it would only produce a bias common to all the tested tecniques. Moreover, for a complete analysis as suggested by the Reviewer, the effect of the machines should be evaluated on the same subjects, and this would introduce several issues including lack of instrumentation, availability of volunteers, clinicians’ time, and violation of the experimental protocol approved by the ethical committee of the Hospital which does not foresee additional examination on the pregnant women.

According to the Reviewer’s comment, we better discussed these aspects in revised version of the manuscript, clearly reporting the study limitation.

Additional remarks:

• Page 2 (line 48, 49) The section relevant to the classes and images in introduction section could be placed in the Fetal PWD dataset section. Also, in the introduction the paragraph “custom dataset of fetal PWD 47 recordings acquired from healthy fetuses at the Division of Pediatric Cardiology, San 48 Michele Hospital, Cagliari, Italy” is redundant and already explained in the “Fetal PWD dataset” section.

>>We thank the Reviewer for this comment. About the last comment, we totally agree and consequently removed the pinpointed sentence. About the first part of the comment, we are not sure about the part the Reviewer is referring to. In our opinion, presenting the images at the beginning is fundamental to understand the target image appearance and its clinical characteristics for people not accustomed with that. Alternatively, this part would be placed too much later in the manuscript. We hope this solution preserves the readability also for the less experienced readers.

• Page 5 (line 130), In the “material and method” section there is no information about the dataset, perhaps the title of this section could be “method”.

>>The dataset description is provided in the last paragraph of the Materials and methods section. We placed it at the end rather than at the beginning of the section to keep the reader focused on the methods.

• For fig 1 provide the extend form for E, A and V (what they are stand for?)

>>We thank the Reviewer and we improved the Introduction section where the E, A and V waves are better explained.

• Please correct the format in equation (1)

>>We thank the Reviewer. Fixed.

• Please add more parameters' definitions in equation (2), what are grandma and grandmin

>>We thank the Reviewer. We better clarify the parameters definition.

• please correct the format in equation (9), × 100 is missing.

>>We thank the Reviewer. Fixed.

• In Performance assessment section, the paragraph “To assess the symmetry of the distributions, we adopted 350 box-and-whiskers plots” please refer to the figure number.

>>We thank the Reviewer. Added.

• Refences in the text are not in order. For example, your first reference is [28] in introduction section.

>>We thank the Reviewer for this comment. We corrected the bibliography.

I look forward to see how the algorithm will perform when test on samples with the different heart conditions.

>> We agree that it would be very interesting to see its performance in this case. It is a development of this work that, however, requires some more years to be completed.

Reviewer #6: In this work, the authors compare between different pulsed wave doppler envelope extraction pipelines by analyzing the different steps involved in the pipeline. They use a downstream classification task accuracy to compare between the signal processing steps involved in the pipeline and recommend important steps in the envelope extraction pipeline.

1. This manuscript introduces very little to no novel methodology and the work is too incremental.

>> Even though little novel methodology is presented, the aim of this study is a systematic analysis of the topic, which is completely lacking in the literature. As also commented by other Reviewers, the classification of fetal beats in PWD traces is an important topic. Several research works speculated on the development of processing chains for PWD envelope tracing introducing little novel methodology and without significant comparison with similar algorithms, just limiting to compare the traced envelope with a single manual tracing. In any case, none of these studies dealed with the automatic identification of the diagnostically meaningful heartbeats, except our previous work (cited). Based on the comments received after the publication of such work, where some colleagues argued that different results could have been obtained by varying the envelope extraction process, we performed a deep literature review and discovered several papers on this topic (far more than those studied for the development of the previous work), so that we decided to perform a systematic analysis on the topic in order to provide a literature review, a fair comparison over a publicly released dataset (no similar works in the field released their data) and a study on the possibility to improve the performance of the algorithms at the state of the art by novel combinations of the processing steps. Based on this, we hope that the Reviewer will see the manuscript from a different perspective.

2. The downstream validation was done on a single classification task using a trained neural network which is insufficient for a lot of claims made in the manuscript. Having supporting validation on multiple downstream detection/classification tasks would add more value to the contribution.

>> We thank the Reviewer for this comment. The aim of this manuscript is the assessment of the different algorithms (and the identification of a new one) for the automatic tracing of the PWD envelope profile finalized at the identification of those fetal beats that are complete and not corrupted by too much noise or by a bad probe positioning so that they can be assessed by the cardiologists, even those with a reduced experience, or automatically measured to provide clinically relevant parameters for the fetal wellbeing monitoring. This task is quite complex by itself and it represents the basis for any tool for the automatic measurement of such parameters in the US devices. This would be very important for the antenatal wellbeing monitoring in clinical settings lacking of pediatric cardiogists accustomed to this kind of complex examination (another option is telecardiology, as presented in Zennaro et al. https://doi.org/10.1371/journal.pone.0164539). For these reasons, we do think the manuscript has a value, as also confirmed by the other reviewers. If the Reviewer is referring to more clinical classification tasks, they should be in any case carried out on the complete and valuable beats, so this work represents an important preliminary step. Our main concern and limitation of the study is the presence of only healthy fetuses. This is a limitation since different morphologies could arise in different pathologic conditions: this aspect deserves further studies, which are not possible with current authorizations from the Ethical Committee according with the agreed protocol. A different protocol is on the way for the the approval but data collection could not start before 2021 and the whole process will last about two years, making it impossible to extend this work with further data.

---

## [Decision Letter · Decision Letter 1]

22 Feb 2021

Impact of pulsed-wave-Doppler velocity-envelope tracing techniques on classification of complete fetal cardiac cycle

PONE-D-20-10354R1

Dear Dr. Sulas,

We’re pleased to inform you that your manuscript has been judged scientifically suitable for publication and will be formally accepted for publication once it meets all outstanding technical requirements.

Kind regards,

Gordon Niall Stevenson, DPhil

Academic Editor

PLOS ONE

Additional Editor Comments (optional):

Reviewers' comments:

Reviewer's Responses to Questions

**Comments to the Author**

1. If the authors have adequately addressed your comments raised in a previous round of review and you feel that this manuscript is now acceptable for publication, you may indicate that here to bypass the “Comments to the Author” section, enter your conflict of interest statement in the “Confidential to Editor” section, and submit your "Accept" recommendation.

Reviewer #1: All comments have been addressed

Reviewer #2: All comments have been addressed

Reviewer #4: All comments have been addressed

Reviewer #5: All comments have been addressed

Reviewer #6: All comments have been addressed

2. Is the manuscript technically sound, and do the data support the conclusions?

Reviewer #1: Yes

Reviewer #2: Yes

Reviewer #4: Yes

Reviewer #5: Yes

Reviewer #6: Yes

3. Has the statistical analysis been performed appropriately and rigorously? 

Reviewer #1: Yes

Reviewer #2: Yes

Reviewer #4: Yes

Reviewer #5: Yes

Reviewer #6: Yes

4. Have the authors made all data underlying the findings in their manuscript fully available?

Reviewer #1: Yes

Reviewer #2: Yes

Reviewer #4: Yes

Reviewer #5: (No Response)

Reviewer #6: Yes

5. Is the manuscript presented in an intelligible fashion and written in standard English?

Reviewer #1: Yes

Reviewer #2: Yes

Reviewer #4: Yes

Reviewer #5: Yes

Reviewer #6: Yes

6. Review Comments to the Author

Reviewer #1: I appreciate the author's responses. I disagree with them in quite a few places, but I don't believe these detract from the manuscript significantly.

For example:

"since the computational complexity of a conventional ANN is dramatically lower than that of a CNN, the former appears to be a better option than the latter for implementation in the US device"

Often a 'simple' neural network with a comparable parameters to a densely connected network outperforms it significantly in image tasks, and all modern mobile phones can, and do, use convolutional neural networks in real time, so I don't think this is a reasonable justification.

I also disagree with the authors assertions that other ultrasound devices will likely only manifest as a bias which can be adjusted for. They have no evidence on which to base this assertion, and time and time again ANN approaches have been shown to generalise egregiously to data in a slightly different format. At least CNNs have the advantages of being relatively spatially invariant due to the use of filters. ANNs don't even offer this advantage. I therefore strongly suggest authors do not make assertions they cannot back up about generalising to different US machines, and instead just acknowledge it as a limitation.

Reviewer #2: (No Response)

Reviewer #4: The reviewer is satisfied that all the general/specific suggestions and comments from the initial review have been addressed adequately by the authors.

Reviewer #5: (No Response)

Reviewer #6: I would like to thank the authors for addressing the comments raised in the previous review. I agree that a systematic analysis and a fair comparison of the envelope extraction process is much necessary for benchmarking developments in PWD processing literature. Even though, very little novel methodology is presented, the systematic analysis presented in this work would be beneficial for the community.

I agree that the classification task presented in this work is an important preliminary step. Even though additional downstream tasks would have added more support and value to the claims in this work, this work is complete (data and analysis) as a first step towards that direction.

This revision with updated text and figures is presented well and acceptable for publication.

7. PLOS authors have the option to publish the peer review history of their article (what does this mean?). If published, this will include your full peer review and any attached files.

Reviewer #1: No

Reviewer #2: No

Reviewer #4: No

Reviewer #5: No

Reviewer #6: No

---

## [Editor Report · Acceptance letter]

6 Apr 2021

PONE-D-20-10354R1

Impact of pulsed-wave-Doppler velocity-envelope tracing techniques on classification of complete fetal cardiac cycles

Dear Dr. Sulas:

I'm pleased to inform you that your manuscript has been deemed suitable for publication in PLOS ONE. Congratulations! Your manuscript is now with our production department.

Kind regards,

on behalf of

Dr. Gordon Niall Stevenson

Academic Editor

PLOS ONE